



# A daily/25km short-latency rainfall product for data scarce regions based on the integration of the GPM IMERG Early Run with multiple satellite soil moisture products

Christian Massari [1], Luca Brocca [1], Thierry Pellarin [2], Gab Abramowitz [3], Paolo Filippucci [1], Luca Ciabatta [1], Viviana Maggioni [4], Yann Kerr [5], and Diego Fernandez Prieto [6]

[1]Via Madonna Alta 126, Perugia, Italy
[2]Univ. Grenoble Alpes, CNRS, IRD, Grenoble INP, IGE, Grenoble F-38000, France
[3]ARC Centre of Excellence for Climate Extremes, UNSW Sydney
[4]George Mason University, Fairfax, VA, United States
[5]Centre d'Etudes Spatiales de la BIOsphère (CESBIO), Université Toulouse 3 CNES CNRS IRD, Toulouse, France
[6]European Space Agency (ESA), Frascati, Italy

**Correspondence:** Christian Massari (christian.massari@irpi.cnr.it)

**Abstract.** Rain gauges are unevenly spaced around the world with extremely low gauge density over developing countries. For instance, in some regions in Africa the gauge density is often less than one station per 10000 $\mathrm{km}^2$. The availability of rainfall data provided by gauges is also not always guaranteed in near real time or with a timeliness suited for agricultural and water resource management applications as gauges are also subject to malfunctions and regulations imposed by national authorities.

5    A potential alternative are satellite-based rainfall estimates, yet comparisons with in-situ data suggest they're often not optimal.

   In this study, we developed a short-latency (i.e., 2-3 days) rainfall product derived from the combination of the Integrated Multi-Satellite Retrievals for GPM early run (IMERG-ER) with multiple satellite soil moisture-based rainfall products derived from ASCAT, SMOS and SMAP L3 satellite soil moisture (SM) retrievals. We tested the performance of this product over four regions characterized by high quality ground-based rainfall datasets (India, Conterminous United States, Australia and Europe)

10   and over data scarce regions in Africa and South America by using Triple Collocation analysis (TC). We found the integration of satellite SM observations with in-situ rainfall observations is very beneficial with improvements of IMERG-ER up to 20% and 40% in terms of correlation and error, respectively, and a generalized enhancement in terms of categorical scores with the integrated product often outperforming reanalysis and ground-based long latency datasets. Given the importance of a reliable and readily available rainfall product for water resource management and agricultural applications over data scarce regions, the

15   developed product can provide a valuable and unique source of rainfall information for these regions.

## 1 Introduction

Rainfall is the main driver of the hydrological cycle (Oki and Kanae, 2006) and plays an essential role in water resource management and agricultural applications (Vintrou et al., 2014; Gibon et al., 2018), drought monitoring (Garreaud et al., 2017) and flood forecasting (Maggioni and Massari, 2018).





Ground networks of rain gauges are considered the most accurate (and as a reflection the most used) rainfall observations across many regions of the world. However, the difficulty and the costs associated with their maintenance along with the timeliness of their data availability are critical obstacles for their use in real-time and seasonal applications. Moreover, while in developed regions the rain gauge distribution is sufficiently dense and supported by well organized and funded organizations,

in developing countries the data coverage is extremely poor.

The number of gauges around the world has been estimated to range between 150000 and 250000 but their distribution is far from being homogeneous (Kidd et al., 2017). For instance, in regions like Africa, South America, and Central Asia the gauge density is often less than one station per $10000 \text{ km}^2$, which results in large interpolation errors of gauge-based gridded rainfall products. This is an interesting paradox, since gauges are insufficient exactly where they are more needed. In these areas the

only source of "observed" rainfall with a timeliness suited for applications is derived from satellite rainfall estimates (SRE) and meteorological models.

SRE are normally derived from sensors on board of and low earth orbiting and geostationary satellites. (LEO; Kidd and Huffman, 2011; Serrat-Capdevila et al., 2014). While geostationary satellites use visible and infrared sensors to retrieve the precipitation signal with a high spatial and temporal resolutions (e.g., 1-3 km, 15-30 min), low earth orbiting satellites use

passive microwave observations to provide global precipitation measurements with a frequency of about two observations per day with a spatial resolutions typically larger than 25 km. The latter are normally more accurate as they provide a more direct measurement of precipitation. A large number of techniques have been developed that exploit the synergy between polar-orbiting retrievals and geostationary observations (Huffman et al., 2007; Hsu et al., 1997; Joyce et al., 2004; Kubota et al., 2007).

The long history of research in the area led in 2014 to the Global Precipitation Measurement (GPM) Mission (Hou et al., 2014), launched by the NASA/JAXA agencies in coordination with the Goddard Earth Sciences Data and Information Services Center (GES DISC). The mission introduced a new concept for rainfall retrieval based on a multi-sensor integration. Within GPM, multiple observations from different instruments are intercalibrated, merged and interpolated with the GPM Combined Core Instrument product to produce half-hourly precipitation estimates on a 0.1° regular grid over the 60° N-S domain through

the Integrated Multi-satellitE Retrievals for GPM (Huffman et al. (2018)). The mission provides three L3 products which are based on different level of timeliness and calibration configurations (the Early run IMERG-ER, the Late run IMERG-LR and the Final run IMERG-FR, see section 2.1.2 for further details).

Although extremely useful, one of the problems with SRE is the instantaneous nature of the measurement, which, along with the intermittent character of the rainfall, make SRE prone to errors (Kucera et al., 2013). For example, precipitation type and

rate (Behrangi and Wen, 2017) along with satellite orbit and swath width (and thus the number of satellite snapshots available) all play an important role in determining the sampling error magnitude (Nijssen and Lettenmaier, 2004; Ciabatta et al., 2017b; Gebremichael and Krajewski, 2004). Other problems are associated with seasonally dependent biases, light rainfall estimation, and detection over snow- and ice-covered surfaces (Ferraro et al., 1994; Ebert et al., 2007; Kidd and Levizzani, 2011; Tian et al., 2007; Gottschalck et al., 2005). Although these problems have been reduced with the advent of the GPM mission thanks



to the a new Dual-frequency Precipitation Radar (DPR), recent works show that there is still room for improvement (Tan et al., 2016; Foelsche et al., 2017; Gebregiorgis et al., 2018b).

Model reanalysis datasets, such as the of European Center Medium Weather Forecast Interim Reanalysis (ERA-interim; extensively described in Dee et al., 2011) and the new ERA5 (European Centre For Medium-Range Weather Forecasts, 2017), are the obvious alternative to ground and satellite-based rainfall products. Although they offer good performance in simulating synoptic weather systems, they often misrepresent the variability of convective systems, mainly due to their relatively low resolution and deficiencies in the parameterization of sub-grid processes (Roads, 2003; Ebert et al., 2007; Kidd et al., 2013; Beck et al., 2017). Although reanalysis datasets perform relatively well globally (Massari et al., 2017) and provide consistent long term precipitation estimation (which is paramount in many research fields) they are normally released with a latency that does not suit water resource and agricultural applications.

Despite these inherent limitations, SRE and reanalysis products are still the only valuable alternative to gauge-based observations within gauge scarce regions and the efforts to improve these datasets by merging procedures or by including other ancillary information has been significantly increasing in the last decade. For instance, Beck et al. (2017) released the Multi-Source Weighted-Ensemble Precipitation (MSWEP), a dataset with a 3-hourly temporal resolution that covers the period 1979 to the near-present. MSWEP is a unique product as it exploits the complementary strengths of gauge-, satellite-, and reanalysis-based data to provide rainfall estimates over the entire globe. Other notable examples are the CHIRPS rainfall estimates (Funk et al., 2015) which are based on a combination of gauges and infrared Cold Cloud Duration (CCD) observations. However, these datasets rely upon the availability of gauge observations, which constitute the "land" or the "bottom up" perspective of the precipitation signal (i.e., the precipitation that effectively reaches the land surface), in contrast to satellite (and reanalysis) estimates, which are more informative about the precipitation in the atmosphere layers (i.e. by cloud and atmospheric models). Where gauges are very sparse or totally missing or their functioning is not guaranteed in near real time, the quality of SRE and models can be significantly affected as the bottom constraint provided by gauges weakens.

A potential solution to circumvent this problem is the use of satellite SM observations as a source of rainfall ground information (Crow et al., 2009, 2011; Pellarin et al., 2008; Brocca et al., 2013; Pellarin et al., 2013; Wanders et al., 2015; Zhan et al., 2015; Ciabatta et al., 2015; Massari et al., 2019). In practice, SM can be used as a trace of precipitation, as the the SM signal after a rain event persists from a few hours to several days. In other words, SM contains information of the amount of water stored in the soil after rainfall. This information can be then exploited to retrieve spatial and temporal characteristics of the precipitation that has effectively reached the land surface. For instance, Brocca et al. (2013, 2014) proposed a direct inversion of the soil water balance equation and used two consecutive satellite SM observations to estimate rainfall fallen within the time interval between the two satellite passes. The underlying idea of this method, known as SM2RAIN, is the use of "soil as a natural rain gauge" as the difference in the water contained in the soil can be directly related to rainfall. This information was used to improve SRE by Ciabatta et al. (2017a) and Massari et al. (2019). Other techniques that exploited SM observations relied upon data assimilation approaches based on sequential filtering techniques, like Kalman filter-based methods (SMART, Crow et al. (2011)) and particle filters (Pellarin et al., 2013; Zhan et al., 2015; Román-Cascón et al., 2017). All of them demonstrated a real benefit for flood forecasting applications (Alvarez-Garreton et al., 2016; Chen et al., 2014; Massari et al., 2018).





In all but two cases (Chen et al., 2014; Tarpanelli et al., 2017), one single SM product was combined with the SRE, a possible limitation if that product does not perform relatively well in the area of interest.

In general, the main advantage of using satellite SM as an indirect measure of ground rainfall information is its uniform temporal and spatial coverage, availability in near real time, and the fact that it transcends national boundaries. Drawbacks are

the low spatial resolution and the relatively low quality in mountainous areas, frozen soils, and dense forests which, however, is also an issue in the case of ground-based observations (due to uneven spatial distribution and data transmission issues in inaccessible areas, undercatch problems, and cost of maintenance). As these problems impact the type of the sensor (active or passive) and the retrieval in different way, their combination would allow to exploit their relative strengths for improving SRE.

In this study, we developed a short-latency (2-3 days depending on the region) rainfall product derived from the combination

of IMERG-ER with multiple satellite SM-based rainfall products. The latter are obtained from the inversion of the SM retrievals derived from: 1) the Soil Moisture Active and Passive (SMAP, Entekhabi et al. (2010)) mission, 2) the Advanced Scatterometer (ASCAT, Wagner et al. (2013)), and 3) the Soil Moisture and Ocean Salinity (SMOS, Kerr et al. (2001)) mission via SM2RAIN. The integrated product is explicitly designed for operational water resource management and agricultural applications over data scarce regions where rainfall observations from hydrometeorological networks are scarce or totally absent.

The integration method we adopted is the Optimal Linear Interpolation (OLI) approach (Bishop and Abramowitz, 2013; Hobeichi et al., 2018) which is based on a technique that provides an analytically optimal linear combination of rainfall products and accounts for both the performance differences and error covariance between the products. We tested the performance of the product 1) over four key regions, namely, India (IN), Conterminous United States (CONUS), Australia (AU) and Europe (EU), where high quality ground-based hydrometeorological networks are available, and 2) in Africa and South America by

using a Triple Collocation (TC) analysis (Stoffelen, 1998). The validity of TC and the consistency of its results with respect to those obtained against classical validation was preliminary tested over the four regions mentioned at point 1 (Massari et al., 2017).

The key strengths of this integrated product are the following:

1. The simultaneous use of multiple satellite SM observations derived from active and passive sensors, which exploits the

advantages of each sensor in improving SRE. Note that ASCAT is on the METOP satellites, which are part of the space segment of the EUMETSAT Polar System (EPS) that will secure the continuation of meteorological observations from the polar orbit in the 2022-2043 timeframe.

2. The short latency (2-3 days, potentially lower in the near future and with L2 products), which is of paramount importance for operational applications like flood forecasting (for medium to large catchments, i.e., >20000 km$^2$), water resource

management, agricultural planning and vector borne disease control.

3. Independence from rain gauge observations, a key factor for data scarce regions like Africa.

The manuscript is divided as follows. Section 2 provides a brief overview of the ground-based and satellite observations used in the study. Section 3 describes algorithms and methods used as well as the integration methodology and the validation strategy. Results are presented in Section 4 followed by discussion and conclusions.



## 2 Data

In this section we describe the datasets used for the integration of IMERG-ER with SM2RAIN rainfall estimates, as well as the datasets used to validate the integrated product.

### 2.1 Regional rainfall datasets

Different ground-based rainfall datasets were used for the four different regions to cross-validate the integrated product, namely, the Australian Water Availability Project (AWAP) in Australia, the ECAD rainfall dataset E-OBS gridded dataset in Europe, the National Centers for Environmental Prediction (NCEP) Stage IV dataset over CONUS and the India Meterological Insititute (IMD) rainfall gridded dataset over India. Below we describe the main features of these datasets (readers interested in more details can refer to the related publications).

1. The Australian Water Availability Project (AWAP) rainfall product is generated via spatial analyses on the quality-controlled daily rain gauge measurements from the Australian Bureau of Meteorology daily rain gauge network. AWAP daily rainfall for a given day is the 24 h total rainfall from local time 9 AM the day before to 9A.M. the current day. The rainfall fields are gridded on a $0.05° \times 0.05°$ grid and spatially resampled to the desired 0.25 degree grid by taking area-weighted averages. Although this product is characterized by a relatively high quality it suffers also from known
shortcomings (the reader interested can refer to (Contractor et al., 2015) for further details).

2. The ECA&D rainfall dataset E-OBS gridded dataset is derived through interpolation of the ECA&D (European Climate Assessment and Data) station data. The station dataset comprises a network of 2316 stations, with the highest station in Northen/Central Europe and lower density in the Mediterranean, Northern Scandinavia and eastern Europe. The E-OBS dataset is derived through a three stage process (Haylock et al., 2008) which bring to different resolutions and grid. In
this analysis, we used the 0.25 degree regular latitude-longitude grid.

3. The National Centers for Environmental Prediction (NCEP) Stage IV (Lin and Mitchell, 2005) is based on the Next-Generation Weather Radar (NEXRAD) measurements, optimally merged with hourly-gauge based observations by using the Multisensor Precipitation Estimator (MPE Seo et al., 2010). This hourly dataset has a spatial resolution of approximately 4 km. The hourly gauge observations in the NCEP Stage IV estimates are derived from the Hydrometeorological
Automated Data System (HADS). Stage IV is characterized by negligible amount (<1%) of missing data over the south eastern CONUS whereas about 90% of the data are missing over the northwest corner of CONUS (roughly between 43°–50°N and 115°–125°). In this study we aggregated the product by averaging all the 4 km pixels falling within the $0.25° \times 0.25°$ footprint. Daily data were obtained by accumulation of hourly observations. In the accumulation procedure, if any missing hourly observation was found in the day, the resulting daily rainfall was discarded.

4. The India Meterological Insititute rainfall gridded dataset is prepared from daily rainfall data of 6955 stations, archived at the National Data Centre, IMD, Pune, by using the Shepard method (Pai et al., 2014). Out of these 6955 stations, 537 stations are the IMD observatory stations, 522 stations are under the Hydro-meteorology program and 70 are Agro





met stations. Remaining stations are rainfall-reporting stations maintained by state governments. The product is released with a $0.25° \times 0.25°$ spatial resolution since 1856.

### 2.1.1 Satellite soil moisture products

In the following we describe the main characteristics of the satellite SM products used in the study. They are:

1. The Advanced SCATterometer (ASCAT) onboard Metop-A and Metop-B and Metop-C satellites is a scatterometer operating at C-band (5.255 GHz) onboard the Metop satellite series. It provides a SM product characterized by a spatial sampling of 12.5 km and from 1 to 2 observations per day depending on the latitude (Wagner et al., 2013). In this study, the SM product provided within the EUMETSAT project (http://hsaf.meteoam.it/) denoted as H115 was used.

    2. The Soil Moisture and Ocean Salinity (SMOS) mission provides a SM product through a radiometer operating at L-band
(1.4 GHz) with  50 km of spatial resolution and 1 observation every 2-3 days (Kerr et al., 2001). In this study, version RE04 (level 3) provided by the Centre Aval de Traitement des Données SMOS (CATDS, https://www.catds.fr/) was used. The version is gridded on the 25km EASEv2 grid and distributed in netcdf format.

    3. SMAP L3. The Soil Moisture Active and Passive (SMAP) mission SM product is obtained by L-band radiometer observations (1.4 GHz) with  36 km and 1/2 observations every three days depending on the location (Entekhabi et al., 2010).
In this study, the version 5 of the Level 3 SM retrievals was used.

    4. AMSR2. The Advanced Microwave Scanning Radiometer 2 (AMSR2) onboard the Global Change Observation Mission for Water satellite is a radiometer operating in the microwave band. Soil moisture retrieval from AMSR2 is obtained from the C and X bands, which allow to obtain a spatial-temporal resolution of  25 km-daily (Kim et al., 2015). In this study, we focused on the X-band SM product obtained by the application of the Land Parameter Retrieval Model to AMSR2
brightness temperature data (Parinussa et al., 2015). Note that, AMSR2 was inverted to obtain rainfall via SM2RAIN but the resulting rainfall was not used in the integration whereas it was used in the validation via TC as an auxiliary dataset.

### 2.1.2 Global Rainfall datasets

In addition to satellite SM products different rainfall datasets were used in the study both for cross comparison purposes and as a part of the integration procedure. In the following the main characteristics of each dataset is provided.

1. The First Guess Daily product provided by the Global Precipitation Climatology Center (GPCC, (Schamm et al., 2014)) is a ground-based rainfall dataset available since 1 January 2009 with a spatial sampling grid of $1°$. This dataset is used wihtin the processing chain of in many gauge-corrected satellite rainfall products. Being based on gauge observations this dataset is very accurate where the station density is relatively high like in Europe, Australia and United States, whereas suffer from serious interpolation errors in areas uncovered by stations. For the sake of comparison, for GPCC
we assumed the same rainfall observed at $1°$ on the $0.25° \times 0.25°$ sub-pixels.



2. ERA5 is the latest climate reanalysis produced by ECMWF, providing hourly data on many atmospheric, land-surface and sea-state parameters together with estimates of uncertainty. The rainfall variable used in this study is characterized by a spatial resolution of  36 km and hourly temporal resolution. ERA5 is available from the Copernicus Climate Change service (https://climate.copernicus.eu/climate-reanalysis). Daily observations of rainfall were computed as the difference

between total precipitation and snowfall. ERA5 was regridded to the the ASCAT grid (25 km) through the nearest neighbouring method to have consistent spatial observations with the satellite SM datasets (see section 3.3).

3. The IMERG algorithm, firstly released in early 2015 (Huffman et al., 2018), is run at 0.1°×0.1° spatial and half-hourly temporal resolutions in three modes, based on latency and accuracy: "early" (IMERG-ER, latency of 4–6 h after observation), "late" (IMERG-LR, 12–18 h), and "final" (IMERG-FR about 3 months). The early and the final runs differentiate in

calibration scheme and in the fact that IMERG-ER has a climatological rain gauge adjustment, whereas the IMERG-FR uses a month-to-month adjustment based on GPCC data.

## 3 Methods

### 3.1 The SM2RAIN algorithm

SM2RAIN (Brocca et al., 2014) is a method of rainfall estimation from SM observations. It is based on the inversion of a 1-

layer water balance equation with appropriate simplifications valid only for liquid precipitation. Assuming a layer characterized by a soil water capacity (soil depth times soil porosity), $Z^*$, the water balance equation can be written as:

$$Z^*ds(t)/dt = p(t) - r(t) - e(t) - g(t) \tag{1}$$

where $s(t)$ is the relative saturation of the soil or relative SM, $t$ is the time and $p(t)$, $r(t)$, $e(t)$ and $g(t)$ are the precipitation, surface runoff, evapotranspiration and drainage rates, respectively. Under unsaturated soil conditions, assuming a negligible evapotranspiration rate during rainfall and Dunnian runoff, solving Eq. (1) for precipitation yields:

$$p(t) = Z^*ds(t)/dt + as(t)^b \tag{2}$$

Note that in Eq. (2) the drainage rate function is of the type $g = as^b$ as in Famiglietti and Wood (1994), with $a$ and $b$ being two fitted model parameters. Once two consecutive SM observations are available and the parameters $a$, $b$ and $Z*$ are known, then Eq. 2 can be used to estimate the rainfall within the time between the two observations. The SM2RAIN parameters $a$, $b$ and $Z*$ are commonly obtained by calibration as described in Ciabatta et al. (2018). For further details on the calibration procedure used within this study the reader is referred to Section 3.3.

### 3.2 The optimal Linear Interpolation approach, OLI

The Optimal Linear Combination approach, OLI (Bishop and Abramowitz, 2013; Hobeichi et al., 2018) provides an analytically optimal linear combination of ensemble members (rainfall estimates in this case) that minimizes the mean square




error when compared to a dataset that is assumed enough accurate to be considered as a calibration dataset $\mathbf{Y_{REF}}$, and thus accounts for both the performance differences and error covariance between the rainfall products. The optimal linear combination is therefore insensitive to the addition of redundant information. This weighing approach has two key advantages: 1) it provides an optimal solution for integrating different rainfall datasets, and 2) it accounts for the error covariance between the different

datasets (caused by the fact that single datasets may share a similar information), that is, they may not provide independent estimates. Given an ensemble of $N + 1$ rainfall estimates and a corresponding calibration dataset $\mathbf{Y_{REF}}$, the weighting builds a linear combination of the $N + 1$ ensemble members that minimises the mean square difference with respect $Y_{REF}$ such that

$$\sum_{j=1}^{N} \left(Y_{REF} - \mathbf{w}^T Y_{PROD}\right)^2 \tag{3}$$

is minimised, where

$$Y_{PROD} = \left[y_{IMERG}, y_{\text{SM2RAIN}_1}, y_{\text{SM2RAIN}_2}, ..., y_{\text{SM2RAIN}_N}\right] \tag{4}$$

represents the different SM2RAIN products plus the IMERG-ER product. The vector of coefficients $\mathbf{w}$ is calculated using

$$\mathbf{w} = \frac{\mathbf{A^{-1}}}{\mathbf{1^T A^{-1} 1}} \tag{5}$$

where $\mathbf{A}$ is the $(N + 1) \times (N + 1)$ error covariance matrix of $Y_{PROD}$ with respect to $Y_{REF}$, and $\mathbf{1^T} = [1,1,1,...,1]$ a vector of $N + 1$ elements. The integrated product is then calculated as:

$$\mathrm{P_{R+SM}} = \mathbf{w^T} Y_{PROD} \tag{6}$$

Note that the OLI method to be analytically optimal requires a bias correction of the ensemble members in $Y_{PROD}$ (i.e., $y_{IMERG}, y_{\text{SM2RAIN}_1}, y_{\text{SM2RAIN}_2}, ..., y_{\text{SM2RAIN}_N}$ in Eq. 4) with the $Y_{REF}$ (i.e., the temporal mean of each member of $Y_{PROD}$ and the mean of $Y_{REF}$ must be equal). In Bishop and Abramowitz (2013) this bias correction was additive, however, for the nature of the precipitation signal (with a considerable amount of null values) a multiplicative bias correction is more appropriate

(Hobeichi et al., 2018). Thus, the latter requires the calculation of appropriate multiplication factors (see section 3.3 for further details).

### 3.3 Integration strategy

This section describes the four steps necessary for obtaining the integrated product $\mathrm{P_{R+SM}}$ (Figure 1). This involves the:

    a) pre-processing of the soil moisture and rainfall products used in the integration (section3.3.1);

b) selection of the parameters of SM2RAIN (section 3.3.2);

    c) selection of the multiplication factors (section 3.3.2);

    d) calculation of the coefficients of OLI via Eq. 5 (section 3.3.3).





Note that a unique calibration dataset, $Y_{REF}$, will be used to perform steps b-d. As $Y_{REF}$ must be characterized by a relatively high accuracy, we performed a preliminary analysis for its proper selection that is described ahead in Section 4.1. Once $Y_{REF}$ is selected it can be used to obtain the coefficients and parameters described at points b-d (i.e., calibration phase, 2015-2017) which can produce integrated rainfall estimates for an independent time period (e.g., 2018 onward) with a latency of 2-3 days.

### 3.3.1 Step 0: soil moisture and rainfall pre-processing

Global SM and rainfall products come with different resolutions and grids. Moreover, the application of the SM2RAIN algorithm to SM observations requires preliminary processing. In Step 0, we resampled all the datasets to the same $0.25°\text{x}0.25°$ grid over land between $\pm 60°$ by using nearest-neighbor interpolation on the ASCAT grid (25 km). In particular, the IMERG products, characterized by a resolution of $0.1°$, were upscaled to $0.25°$ using a box-shaped kernel with antialiasing, an approach that was found to outperform simple spatial averaging. Rainfall accumulations were aggregated to daily-scale (from 00:00 to 23:59 UTC).

As satellite SM data are not provided regularly spaced in time and contain gaps (for instance we did not include in the analysis observations characterized by frozen soils, snow presence, or radio interference contamination, by using the specific flags for each product), they were linearly interpolated at 00:00 UTC to produce SM2RAIN daily rainfall from 00:00 to 23:59 UTC (see step 1). Note that we limited the interpolation to a maximum of 2 days; beyond that we assumed SM2RAIN rainfall as missing (in these cases only IMERG-ER is used in the integrated product as better described in section 3.3.3). Note that the amount of missing data is generally dependent upon the location. Locations where the quality of satellite SM observations is poor are characterized by a lot of missing data and the integrated product is basically close to IMERG-ER.

### 3.3.2 Step 1 and 2: calibration phase

Step 1 refers to the calibration of SM2RAIN for the selection of the optimal parameters distribution pixel per pixel. All the parameters described in section 3.1 were selected by using $Y_{REF}$. For that, we minimised the daily root mean squared error (RMSE) between the SM2RAIN rainfall applied to the specific SM product and $Y_{REF}$ during 2015-2017. The products so obtained are referred to SM2RAIN-ASCAT if the satellite SM observations were derived from ASCAT, SM2RAIN-SMOS if derived from SMOS, and SM2RAIN-SMAP if derived from SMAP. In addition to these products, we also produced SM2RAIN-ASMR2* and SM2RAIN-ASCAT*, by using satellite SM observations derived from AMSR2 and ASCAT with non-calibrated parameters, i.e., we used constant parameters globally derived from previous studies as in Massari et al. (2017). Remember that, these two last products were not used in the within OLI but will serve then only for validation purposes with TC.

As depicted in Section 3.2, the application of OLI requires unbiased ensemble members. This implies matching the long term temporal mean of $Y_{REF}$ with the ones of IMERG-ER, SM2RAIN-ASCAT, SM2RAIN-SMOS and SM2RAIN-SMAP by using a different (and temporally constant) multiplication factor for each member (i.e., a factor that multiplied by the mean of the member guarantees the matching with the mean of the calibration dataset). However, applying this procedure resulted in an overall reduction of the quality of the SM2RAIN members because a temporally constant multiplication factor deteriorated the quality of light rainfall (<5 mm/day) with an increase of the false alarms (due to the noise contained in the satellite SM time





series). To overcome this issue, we adopted a slight different strategy which, despite does not guarantee a perfect matching of the long temporal means and thus is not theoretically optimal, limited the problem of the increase of false alarms. In practice, for each member (i.e., SM2RAIN-ASCAT, SM2RAIN-SMOS, SM2RAIN-ASCAT,SM2RAIN-SMAP and IMERG-ER) we calculated the the ratio between its mean monthly rainfall (i.e., mean of all the Januaries, mean of all the Februaries and so on)

and the monthly mean of $Y_{REF}$ (obtaining one multiplication factor per month per pixel for a total of 12 multiplication factors for each grid point). These factors were then used to multiply the daily rainfall observations of each member (relative to the desired month) to obtain a monthly-based rescaled daily rainfall estimate.

This procedure is in principle a climatological correction rather than a bias correction because it uses the climatology of $Y_{REF}$ as a reference. It guarantees a more consistent spatial pattern of rainfall among the members prior to the application of

OLI which helps also to avoid spatial inconsistencies when different combinations of members are used within the integrated product. Note that this operation does not constrain the variability of the precipitation from year to year to the one of $Y_{REF}$ as it only redistributes rainfall within the year and guarantees all the members to be realigned to the same climatology. Note also that a similar procedure is used for the production of IMERG-ER and IMERG-LR products (Huffman et al., 2018), and can be easily implemented for its use in near real time once the 12 factors for each member are known. From here onward we will

refer to this procedure as a climatological correction.

### 3.3.3 Step 3: application of OLI

For the application of OLI (i.e., integration) we proceeded by considering these three methodological aspects:

1. First off, we performed a quality check, by comparing the correlation coefficient of each SM2RAIN product with the calibration dataset ($Y_{REF}$). When the correlation was found less than 0.4 (i.e., no correlation), the product was automat-

ically excluded and the OLI was applied on the reminder of them. If all the SM2RAIN products correlation fall below this threshold (for example in dense forests or high mountainous regions), only IMERG-ER was retained.

2. The calculation of the OLI coefficients in Eq. 5 is not computationally demanding and uses the full calibration time series (2015-2017). In particular, Eq. 5 provides the specific coeffiecients to be used in Eq. 6 at each time step. If one of the SM2RAIN product is not available at a specific time step for the reasons described in Step 0 (see section 3.3.1),

we linearly redistributed the coefficients to the products available at that time step so that their sum is one (to ensure unbiased estimates).

3. The application of OLI among the SM2RAIN products and IMERG-ER was carried only when IMERG-ER values are larger than zero taking advantage of the enhanced rain/no rain detection accuracy of IMERG that uses DPR (Gebregiorgis et al., 2018a) whereas when IMERG-ER was zero this value was kept in the merged product. This tactic mitigates the

degradation of rainfall estimates during low-rainfall time steps as demonstrated by Massari et al. (2019).





4.  The final product is then composed of multiple rainfall datasets weighed according to Eq. 6. IMERG-ER is always present whereas the presence of the three SM2RAIN rainfall estimates derived from ASCAT, SMOS, and SMAP depends on their relative accuracy (if they satisfies the threshold) and availability in time and space.

The success of the overall procedure described above is dependent upon the quality of $Y_{REF}$. Although the calibration phase

seems very intensive, it will be demonstrated in Section 4 that, if $Y_{REF}$ has a relatively good accuracy, its effect on the final quality of the integrated product is very low. However, its choice is strategic in some regions, as will be shown in section 4.1, and thus deserves a careful investigation.

### 3.4  Validation strategies

For the validation of the integrated product two different strategies were followed. First, we selected four key regions charac-

terized by different climates and landscapes (i.e., CONUS, AU, EU and IN) where ground-based observations (derived from rain gauges and rain gauges plus radar) are very dense and of high quality (see section 2.1). Both continuous and categorical scores are considered, as commonly used in a classical validation of global precipitation products (see Maggioni et al. (2016) for further details).

Next, since many areas of the world like Africa, South America, and Central Asia have highly variable density of rain

gauges, validation was also performed using a TC analysis as proposed by (Massari et al., 2017; Khan and Maggioni, 2019) (see section 3.4.2). TC offers a viable way to validate rainfall products in data scarce regions by providing (theoretical) error and correlation of each product with the "unknown" truth. Note that, we tested the validity of the TC validation by applying it to the same key regions where the classical validation was carried out. Then, TC was applied to Africa and South America to validate the integrated product and the other datasets that are part of the analysis. The validation with TC was carried in 2018

(only one year) which is independent from the calibration period (2015-2017).

### 3.4.1  Classical validation

Continuous and categorical error metrics were both adopted for validating daily rainfall. The continuous scores were the Pearson correlation coefficient (R), the Root Mean Squared Error (RMSE), and the additive bias (BIAS). In addition, three categorical scores were considered: the Probability of detection POD=H/(H+M), which measures the likelihood of the rainfall

estimate to detect an event when it in fact occurs; the False alarm ratio FAR = F/(F+H), which measures the likelihood that a precipitation event does occur when reference does not estimate rain; and the Threat Score (TS), which is an integrated measure of POD and FAR. All these scores are based on the contingency Table 1. In the table, H represents hit cases, when both the precipitation estimate and reference are greater than or equal to the rain/no-rain threshold percentile ($th$); F represents false alarms, when precipitation estimate is greater than or equal to $th$, but the reference is less than $th$; M represents missed events,

when the reference is greater than or equal to $th$, but the precipitation estimate is less than $th$; and Z represents correct no-rain detection, when both the precipitation estimate and reference are less than $th$. N is the sample size, i.e., the total number of observed events and $N = H + M + F + Z$.





### 3.4.2 Triple Collocation Analysis applied to rainfall observations

In this study, TC analysis (Stoffelen, 1998) was applied to estimate the correlation and the error of the rainfall estimates when a reliable reference is missing like in Africa. Here we present a summary of the theory behind TC while the reader interested in more details can refer to Massari et al. (2017).

Suppose we have three measurement systems $X_i$, observing the true variable $t$ characterized by an additive error model

$$X_i = X_i' + \varepsilon_i = \alpha_i + \beta_i t + \varepsilon_i \tag{7}$$

where $X_i$ ($i =$1, 2, 3) are collocated measurement systems linearly related to the true underlying value $t$ with additive random errors $\varepsilon_i$, respectively, while $\alpha_i$ and $\beta_i$ are the ordinary least squares intercepts and slopes. Assuming that the errors from the independent sources have zero mean ($E(\varepsilon_i)$=0) and are uncorrelated with each other ($Cov(\varepsilon_i, \varepsilon_j)$ =0, with $i \neq j$) and with $t$ ($Cov(\varepsilon_i, t)$=0) the variance of the error of each dataset can be expressed as ((McColl et al., 2014)):

$$\sigma_\varepsilon = \begin{bmatrix} \sqrt{Q_{11} - \frac{Q_{12}Q_{13}}{Q_{23}}} \\ \sqrt{Q_{22} - \frac{Q_{12}Q_{23}}{Q_{13}}} \\ \sqrt{Q_{33} - \frac{Q_{12}Q_{23}}{Q_{12}}} \end{bmatrix} \tag{8}$$

where $Q_{ij} = Cov(X_i, X_j)$ is the covariance within the variables $X_i$.

In addition, McColl et al. (2014), using the definition of the correlation and covariance, demonstrated that:

$$R^2_{TC(\mathbf{t},\mathbf{X})} = \begin{bmatrix} \frac{Q_{12}Q_{13}}{Q_{11}Q_{23}} \\ \frac{Q_{12}Q_{23}}{Q_{13}Q_{22}} \\ \frac{Q_{13}Q_{23}}{Q_{12}Q_{33}} \end{bmatrix} \tag{9}$$

where $R^2_{TC(\mathbf{t},\mathbf{X})}$ is the squared correlation coefficient between $t$ and $X_i$ (McColl et al., 2014).

Note that the error (and correlation) calculated via TC are generally lower (higher) than those calculated using the classical validation given that it does not include the reference uncertainty.

### 3.4.3 Validation mask

Although the integrated product is potentially available everywhere, we found that where the quality of satellite SM observations is very low like in forests, frozen soils and mountainous areas OLI coefficients associated to the SM2RAIN products was very small and the integrated product was mainly constituted by IMERG-ER. Therefore, to avoid any misinterpretation about the real benefit of integrating IMERG-ER with satellite SM observations we limited the validation of the integrated product to

the ASCAT committed area (Hahn, 2016). The area is limited to low and moderate vegetation regimes, unfrozen and no snow cover, low to moderate topographic variations, no wetlands, no coastal areas, no deserts (see Figure A1 of the supplementary information). Outside of this area, satellite SM observations might suffer from several problems and are weighed much less by OLI (although we also found benefits here, see sections 4.1.1 and 4.1.1). In addition, for the sake of product distribution and use we can ensure optimal results only over this area and thus we associated a flag to the pixels fallen outside it in the netCDF

file included in the supplementary information.





## 4 Results and discussion

Both the calibration of SM2RAIN and the OLI implementation need a calibration dataset as described in Section 3.3 (i.e., $Y_{REF}$). The choice of this dataset is strategic for obtaining a good quality integrated product. Section 4.1 describes the process of the selection of $Y_{REF}$ considering different potential candidates. Sections 4.1.1 and 4.1.3 describe the validation over US, IN, AU and EU by using the hydrometeorological networks described in Section 2.1 and the validation in Africa and South America by using TC (section 3.4.2), respectively.

### 4.1 Calibration dataset selection

The choice of a calibration dataset is strategic for both the SM2RAIN parameters selection and the OLI coefficients calculation. Thus, it has to be carefully selected based on i) accuracy (i.e., low error and high correlation with "true" rainfall), ii) homogeneous performance in time and space, and iii) continuous spatial and temporal coverage (as well as spatial and temporal resolution closer to the one of the rainfall to be estimated). Potential candidates are:

1. GPCC, which has potentially high accuracy and low biases where the rain gauge coverage is good, but can be unreliable when the rain gauge distribution is scarce (e.g., Africa and South America). It might also suffer from time dependence performance as a function of rain gauge availability.

2. ERA5, which provides full coverage and generally homogeneous performance all over the world.

3. IMERG-FR, which is a gauge-corrected satellite product and potentially highly accurate where rain gauges distribution is dense. The drawback is highly dependent upon IMERG-ER where rain gauge observations are scarce. For this reason, we initially excluded this product from the list of potential candidates and focused only on the other two.

To explore the performance of ERA5 and GPCC we applied TC as described in 3.4.2 during the whole period 2015-2018. The following triplets were used (by keeping in mind the need to satisfy the assumptions of TC highlighted above):

1. GPCC, IMERG-ER, and SM2RAIN-ASCAT* (triplet A)

2. GPCC, IMERG-ER, and ERA5 (triplet B)

3. GPCC, ERA5, and SM2RAIN-ASCAT* (triplet C)

Note that SM2RAIN-ASCAT* above is not the one used in the integration, but it was produced using constant parameters $a$, $b$, and $Z$ all over the world (i.e., it is not regionally calibrated) as in (Massari et al., 2017) to avoid potential violation of the TC assumptions. On the other hand, even using SM2RAIN-ASCAT (the calibrated dataset), similar results were obtained (not shown) as we found a negligible effect of the calibration on TC results (in terms of TC correlations).

Table 2 shows results for triplet A, B, and C. Different configurations of the triplets provide similar results, suggesting that TC can be considered reliable. In particular, ERA5 performs the best among all. Figure 2a plots the number of stations used for the GPCC First Guess 1.0 degree product for years 2015-2018 whereas Figures 2b and 2c show the TC temporal correlation of





the GPCC and that one of ERA5 for the period 2015-2018. An interesting feature is that lower correlations of GPCC closely match with low station density areas (by comparing Figures 2a and 2b) whereas ERA5 shows a more homogeneous and higher correlation overall the globe except a fall of the performance in the Sahel region where convective type of systems are very common (which is a well known issue as described in Section 1). It has also to be noted that the number of stations used

by GPCC in Africa during this period are very low with areas totally uncovered which likely lead to significant interpolation error. The uneven rain gauge spatial distribution seems to significantly impact the GPCC quality and in turn can potentially cause sub-optimal performances, if used as a calibration dataset. ERA5 relies less on observation density and shows a more homogeneous performance pattern with respect to GPCC. Thus, ERA5 was selected as $Y_{REF}$.

Note that, based on this choice, the integrated product is totally independent upon rain gauges. This allows to independently

cross-validate the integrated product in EU, IN, AU and CONUS during 2015-2017 against high quality ground-based rainfall observations (see section 4.1.1). The latter serves to understand if the entire procedure of integration described in section 3.3 is correct and provide an overall idea of the maximum potential performance that can be obtained by the integrated product (being performed in the same period used for calibration).

### 4.1.1   Classical validation over key regions using high-quality ground-based observations

Figure 3 summarizes the products used in $P_{R+SM}$ pixel per pixel for the different key regions. While for India and Australia, SM2RAIN-ASCAT, SM2RAIN-SMOS, and SM2RAIN-SMAP are present almost everywhere, over CONUS and Europe there exist areas where SM2RAIN-SMOS was not used either because too high Radio Frequency Intereference was found in the SMOS product or because of its relatively low performance (Chen et al., 2018). In the figure, we did not superimpose the mask described in section 3.4.3 to show that the areas in dark blue (i.e., where only IMERG-ER is retained) almost coincide with

the ASCAT-committed area. For instance the north-eastern CONUS region is known to be a challenging area for satellite SM products and, as a result, here $P_{R+SM}$ relies on IMERG-ER alone. Similarly, the coastal areas are mostly characterized by dark blues pixels, which indicates no integration with any SM2RAIN product. Note however that the ASCAT committed area does not always match the area where only IMERG-ER is present, e.g., north eastern Africa. Here, ASCAT SM product is known to perform relatively bad due to volume scattering (Wagner et al., 2013), whereas passive products perform relatively

well (in orange the presence of only passive sensors integrated with IMERG-ER can be seen). Although in areas like this we still have an improvement of IMERG-ER and thus they could be considered part of the integrated product we preferred to be conservative and guarantee the product reliability only over the mask described in section 3.4.3.

Table 3 shows R, RMSE, and daily bias of the different rainfall products obtained by using the ground-based observations described in Section 2.1 as references. The short latency products are shown (less than 2-3 days) in light blue, in contrast to the

long latency ones (larger than 2 months). The integrated product $P_{R+SM}$ significantly outperforms IMERG-ER, indicating the benefit of including SM information in IMERG-ER via OLI. In addition, $P_{R+SM}$ is slightly better than IMERG-FR and in three out of four regions better than ERA5, which was used as calibration dataset. This suggests that the selection of the calibration dataset is not necessarily a major limiting factor in the proposed framework, as satellite SM contains inherent information about rainfall, as long as its quality is sufficiently high. When compared with long latency products, GPCC is particularly





good in Europe and Australia, which is expected due to the large amount of gauge stations shared with the references. RMSE results show the relatively good performance of the integrated product with best performances in India. Larger errors were observed for IMERG-ER and smaller errors for GPCC for the same reasons mentioned above. The integrated product has similar performance to ERA5, as expected.

Figure 4 shows, for AU, the median increase in temporal correlation (2015-2017) with respect to IMERG-ER obtained by integrating the latter with one (either ASCAT or SMAP or SMOS), two (either ASCAT+SMOS, or SMAP+SMOS or ASCAT+SMAP) or three SM2RAIN products (ASCAT+SMAP+SMOS). The addition of multiple products, though beneficial, gets smaller as we ingest more SM2RAIN-based rainfall estimates. This is due to the redundancy of information provided by SM, which causes no further improvement. Although this might suggest that using a single SM2RAIN product is equivalent to

using multiple products, the use of multiple products always guarantees optimal performance per pixel and is useful where one of the products does not perform well, as shown Figure 3. Results for the other key regions provide similar overall conclusions and are not shown here.

Figure 5 shows the correlation and RMSE differences in percentage obtained between the integrated product and IMERG-ER. Blue areas are those characterized by improvements, whereas red denotes deterioration. There is an overall improvement

for both scores over the study areas. Larger improvements are obtained in terms of RMSE, which in some cases (i.e., CONUS) are larger than 40 %. In terms of R, the improvement spans from 5 to 15% with larger values obtained for Europe and Australia. There are also spots over northwestern CONUS characterized by deterioration. We attributed this to the low agreement between stage IV data and ERA5 data (used as calibration dataset), which can be also found in (Beck et al., 2019). Note this is a challenging area for Stage IV data as also demonstrated by Tian et al. (2007) who found significant performance differences of

the 3B42 rainfall product in northwestern CONUS when compared either to the CPC Unified Gauge-based Analysis of Global Daily Precipitation (Higgins et al., 2000) product or with the Stage IV dataset.

To understand the benefit of integrating SM-based rainfall with IMERG-ER as a function of the topographic complexity, Figure 6 shows the median differences, in terms of correlation (panel a) and RMSE (panel c), obtained by $P_{R+SM}$ with respect to IMERG-ER for CONUS. The topographic complexity come along the ASCAT H115 product and is computed as

the normalized standard deviation of elevation using GTOPO30 data (Hahn, 2016). It ranges between 0% for flat areas to 100 % for very complex terrain. The integrated product is able to improve the quality of IMERG-ER over flat areas better than complex terrain. This result is somehow expected, as we know that the topographic complexity impacts the quality of the SM retrieval.

The benefit of the integration was also computed as a function of land cover (panels b and d in Figure 6 for CONUS). Land

cover information comes from the ECOCLIMAP dataset (Champeaux et al., 2005), provided at 1km spatial resolution. We have simplified the original land use classes into 8 categories: bare land, rocks, urban, forest, wooded grassland, shrub land, grassland, and crop. Except for urban, rock, and bare soil (with a percentage of pixels within CONUS less than 0.5%), the integrated product performs better over shrub land, grassland, and crop, whereas lower performance are obtained over forests. This result is also expected as the quality of the satellite SM product can be highly impacted by the presence of dense vegetation

for the difficulty of the retrieval in separating the effect of the soil water content from the water contained in leaves.





Figure 6 refers to CONUS as we found highly representative of different landscape complexity and land cover type. Results for AU, EU and IN picture very similar findings and are reported in the supplementary information (Figures A2 and A3)

Figure 7 shows the differences in terms of POD, FAR and TS between the integrated product $P_{R+MS}$ and IMERG-ER as a function of the rainfall percentiles. As the correction of IMERG-ER was only carried for positive rainfall values and

SM2RAIN-based rainfall lower than 1 mm was assumed unreliable (to exclude the possibility to interpret satellite SM noise as rainfall Zhan et al. (2015)) the differences with respect to IMERG-ER are visible only above the 50-60$^{th}$ percentiles. After 50-60$^{th}$, a significant increment of POD is evident for all the study regions, whereas the difference in FAR denotes a slight deterioration for smaller rainfall accumulations. The improvement in terms of FAR becomes significant for higher rainfall accumulations. The overall improvements is shown by the TS score, which is generally positive suggesting that the integrated

product helps to improve IMERG-ER in terms of categorical scores especially for 70-90$^{th}$ percentiles.

### 4.1.2   Assessment of the validity of the TC analysis

Here we test whether TC is representative of the results obtained using classical validation. TC provides error and correlation with an unknown truth, as a result the TC scores are generally more optimistic than those obtained with classical validation. To calculate TC correlations and errors, we built a total of 15 triplets by combining the ground-reference described in section

2.1, IMERG-ER, IMERG-FR, ERA5, GPCC, $P_{R+SM}$, SM2RAIN-AMSR2*, and SM2RAIN-ASCAT* (the last two are not calibrated datasets with global constant parameters ) in different ways. We obtained a distribution of errors and correlations derived from using multiple triplets for the same product (i.e., summarized by a box plot). The underlying idea is that, if the choice of the triplet is correct, these errors and correlations should not differ too much for the same product from triplet to triplet and should provide similar conclusions to the classical validation analysis.

Moreover, we performed the TC analysis with two configurations. In the first, we included in the same triplet products that would apparently violate the TC assumptions (we only excluded those triplets where the violation is obvious, i.e., like GPCC with the ground-based reference). Specifically, we used triplets containing $P_{R+SM}$ and ERA5 (i.e., the calibration dataset), as shown in Table A1 of the supplementary information. This configuration allows investigating the existence of a possible "false" beneficial increase in performance for the integrated product when it is used in conjunction with ERA5 (i.e.,

the calibration dataset) in the same triplet. In the second configuration, we performed the same TC analysis by removing those triplets containing ERA5 with the integrated product (Table A2). Here, we only kept triplets containing SM2RAIN-AMSR2* combined with $P_{R+SM}$. Note that SM2RAIN-AMSR2* was not used in the integration and was not calibrated with ERA5.

Results show that the differences between the two configurations are negligible (compare Figure 8 for the first configuration for AU with Figure A5, panels a and b related to AU for the second configuration). This allows assuming a negligible effect of

the calibration dataset when used in the same triplet with the integrated product. However, in the second configuration, the risk of having cross-correlated error dependence is not null, as the SM2RAIN algorithm was used both in the integrated product and in SM2RAIN-AMSR2*. To check this possibility, we calculated these spurious error cross-correlations by assuming the ground-based observations for the key regions (i.e., AWAP, Stage IV, E-OBS and IMD) as a true rainfall and found them always below 0.4. This provides an additional proof about the validity of the TC analysis.





The ability of TC to provide similar conclusions of a classical validation analysis is discussed in Figure 8. The figure shows the box plots obtained by considering TC errors and correlations of Table A1 for AU. The small spread around the median value suggests consistent results among the different triplets. In addition, the relative quality of the products provided by the classical validation (i.e., red dots are the median value obtained via classical validation) seems reasonably reproduced by TC

both in case of correlation and error. They are generally lower (higher for the error) with respect to TC results as they contain already the uncertainty of the reference (i.e., AWAP). In the case of GPCC, the values are closer or higher to the one obtained via TC due to the obvious agreement with AWAP (which shares a significant number of rain gauges with GPCC). Moreover, AWAP is characterized by a relatively high quality, which justifies its for validation purposes. Results for the other study regions (CONUS, EU, and IN) in Figures A4 and A5 show similar conclusions suggesting that TC is a reliable technique to

validate rainfall products.

### 4.1.3 Validation over data scarce regions using TC

Figure 9 shows the product combinations for each pixel of the study areas used for obtaining $P_{R+SM}$ in Africa and South America. These combinations and the associated OLI coefficients (including the SM2RAIN parameter calibration) were obtained during the calibration period 2015-2017. Areas where all SM2RAIN products are ingested match with those characterized by

relatively good quality of satellite SM observations, i.e., those not characterized by dense forests, desert areas and frozen soil, as well as snow covered areas. This suggests that the integration is robust and exclude meaningfully low-quality SM information.

Unlike the results presented in Section 4.1.1, here we validate the products during 2018, independent from the calibration period (i.e., 2015-2017). As in Africa and South America the rain gauge distribution is scarce (see Figure 2a), the validation was carried out via TC, using 3 triplets built among ERA5, SM2RAIN-ASCAT*, $P_{R+SM}$, GPCC, and IMERG-ER:

1. ERA5-GPCC-IMERG-ER

    2. ERA5-GPCC-SM2RAIN-ASCAT*

    3. ERA5-GPCC-$P_{R+SM}$

Figure 10 shows $R^2_{TC}$ (left) and TC-RMSE (right) over Africa obtained by triplets 1 (ERA5-GPCC-IMERG-ER) and 3 (ERA5-GPCC-$P_{R+SM}$). In particular, panels a-d refer to the short latency products while the rest of them are long latency ones

(>2 months). The integrated product outperforms both IMERG-ER and long-latency products like GPCC and ERA5 as we found in section 4.1.1. ERA5 is characterized by lower performance in the Sahel region as highlighted in Figure 2b whereas GPCC is strongly affected by the uneven rain gauge distribution as depicted in Figure 2c. Similar results are obtained for South America in Figure 11, where the central eastern part gets greener (higher correlation) and whiter (lower error) after integration with SM2RAIN based rainfall estimates. In South America the performance of ERA5 is higher than the one obtained in Africa

and consistently more homogeneous.

Figure 12 summarizes the results obtained in the two regions by considering only the committed area (panels a and b) and all the pixels of the analysis (non-masked by the committed area, panels c and d) also in terms of boxplots. It can be seen that





in Africa (panels a and c) the integrated product is always the best both in terms of error and correlation. In South America (panels b and d) ERA5 outperforms the integrated product if no mask is used (panel d). A reason for that is the lower skill of IMERG-ER over dense forests especially in terms of error, which impacts the overall quality of the integrated product. In particular, relatively good performance are obtained in Africa over the Sahel region and in South America over eastern Brazil.

## 5  Discussion and conclusions

In this study, we have developed a procedure to obtain a short latency (less than 2-3 days), 25km/daily satellite-based rainfall product based on the integration of IMERG-ER with SM2RAIN-based rainfall estimates derived from three different satellite SM products (i.e., SMOS, SMAP and ASCAT). With this latency – potentially reduced to about 1 day via the use of L2 products – the product targets agricultural and water resource management applications over data scarce regions like Africa, South America and Central Asia.

To merge SM2RAIN based rainfall estimates with IMERG-ER, we used the OLI approach previously used by Bishop and Abramowitz (2013) to combine different climate model estimates. The procedure optimally merges multiple estimates of the same variable by minimizing the error with a calibration dataset. The choice of this calibration dataset was discussed and analyzed in detail by applying Triple Collocation analysis to different candidates leading to the choice of ERA5 reanalysis rainfall product. In the procedure, no gauge information was directly used either in the calibration of SM2RAIN or in the integration of estimates via OLI, therefore the developed product is totally independent from ground-based observations of rainfall (except the inherent gauge information contained in IMERG-ER).

The integrated product was cross-validated with high quality ground-based rainfall observations in Australia, India, Europe, and continental United States and cross-compared in the same regions against long-latency products (i.e., released with a time span of 1-2 months and thus not suited for operational applications). The validation entailed different continuous and categorical scores and was carried out for different land cover classes and as a function of the topographic complexity. In this respect, we found:

1. the integrated product performed relatively well and often better than the long latency products, which are designed to obtain best performance as they ingest many observations and use gauges (often the same used here for validation). The best product in regions with high density rain gauge observations was found to be GPCC (although this product is obviously correlated with the ground reference). An interesting feature was the better performance of the integrated product with respect to the calibration dataset which highlights the high value of information provided by SM. These results are relevant given that the integrated product can be potentially released within 2-3 days.

2. the improvement of IMERG-ER was relevant and ranged from 10-15 % in terms of correlation and up to 40 % in terms of RMSE in some cases. A smaller impact was obtained over very dense forests and complex terrain given the inherent limitations of satellite-based observations over these areas. We also observed deterioration in correlation in some areas of northwestern CONUS and India which need further analysis.





3. An additional validation, totally independent from the calibration, was carried out in Africa and South America. Here, due to the lack of a reliable benchmark dataset, we adopted TC analysis (after having validated it) to calculate error and correlation of the integrated product, IMERG-ER, GPCC as well as ERA5. Results confirm those obtained via classical validation with the integrated product outperforming IMERG-ER. Moreover, in data scarce regions, the integrated product outperforms GPCC and providing similar performance to ERA5 (better in the Sahel region).

Despite the good performance achieved by the product several aspects need further investigation.

1. The short time records of some of the satellite-based observations used in the integration (i.e., SMAP and IMERG-ER) limited the length of the calibration period which could impact the calculation of the climatological correction procedure and the OLI coefficients shown in the method section. It also shrinks the length of the validation period which was restricted to 2018. Future work will focus on further validation of the products by extending the validation to 2019 and beyond.

2. Although TC is a possible (and likely the only) alternative for evaluating rainfall estimates over data scarce regions, it does not provide a thorough evaluation of the rainfall estimates as it does not provide information about categorical scores and bias. Therefore, over these regions it is not guaranteed that the integrated product performance is optimal in this respect. Future work should focus in testing the product for applications like flood prediction, water resource management, crop modelling and risk insurance. Note that first attempts in using the product for flood prediction (not shown in this study) are providing promising results.

3. The integration is not possible everywhere given the low quality of the satellite SM observations over dense forests and the lack of SM information over frozen surfaces. We can only have confidence in the optimal performance of the integrated product over the area described in section 3.4.3. This of course does not exclude the possibility that the product might work well outside of this area.

4. Daily/25 km temporal/spatial sampling might be not adequate for small scale applications. Future work should therefore take into account satellite SM products with higher spatial resolution (e.g. Piles et al., 2011; Merlin et al., 2012; Malbéteau et al., 2016; Bauer-Marschallinger et al., 2018a, b; Chan et al., 2018) and shorter revisit times. Note that with the current constellation of Metop A, B and C satellites in addition to the future SCA scatterometer (Rostan et al., 2016), or with potential availability of geosynchronous C-band radars we will have the opportunity to collect multiple satellite SM observations within the day which could be used to calculate sub-daily rainfall estimates from SM observations.

5. Despite 2-3 days of latency being fine for many applications, it might not be sufficient for rainfall monitoring in real time and flood forecasting in medium to small basins. In this respect, IMERG-ER, with its 4-5 hours of latency is the only satellite product potentially providing rainfall observations that could be used for such applications, although in that case not only the latency is important but also the spatial resolution. Future work should focus in the integration of L2 satellite SM products with IMERG-ER also using alternative integration schemes and products with respect to those used in this study.





6. The record length of the product is restricted to the GPM and SMAP eras (i.e., 2015 onward). This potentially limits the use of the products for drought and flood frequency analysis. However, the integration procedure does not rely upon the availability of the above products but can be applied to any other long-term rainfall and soil moisture dataset available. Note that all the IMERG data will eventually be retrospectively processed to the start of the TRMM era (from March 2000 to present) and SM observations are available back to 1978 ((Dorigo et al., 2017)). Therefore, there is a large potential for developing a long-term integrated product specifically target for climate applications.

*Data availability.* $P_{R+SM}$ is available via https://zenodo.org/record/3345323.XThcfHvOOUk



*Competing interests.* The authors declare no conflicts of interest.

*Acknowledgements.* This work is supported by the European Space Agency ESA (contract 4000114738/15/I-SBo) project SMOS+Rainfall Land II. Gab Abramowitz acknowledges the support of the Australian Research Council Centre of Excellence for Climate Extremes (CE170100023).





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





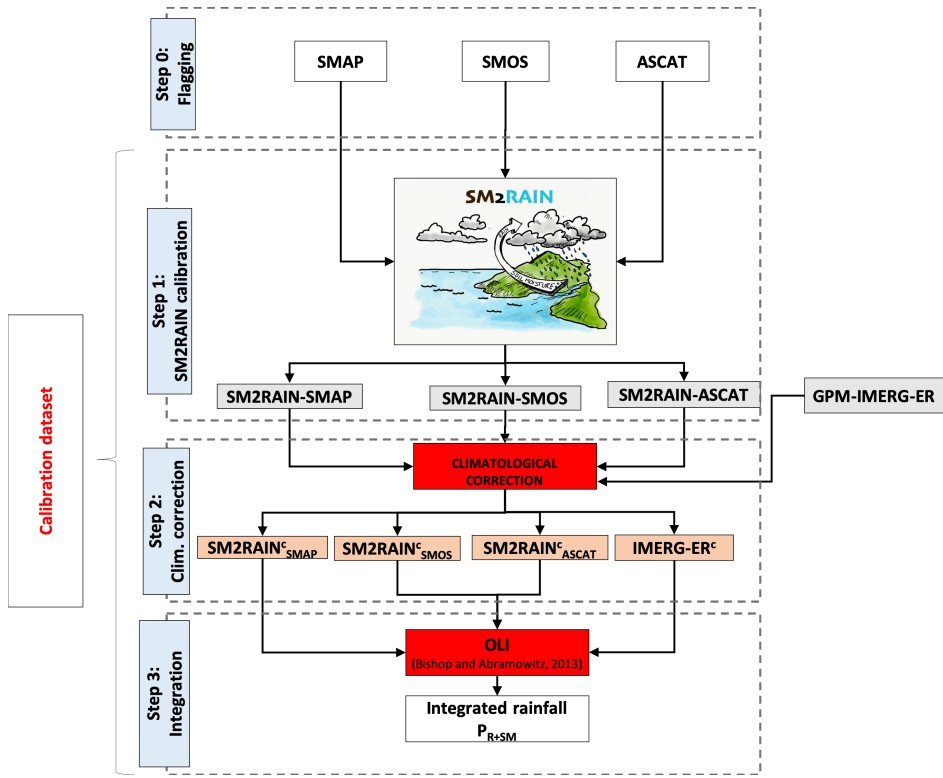

**Figure 1.** Integration scheme used for the calculation of the integrated rainfall product PR+SM. In step 2 $\mathrm{SM2RAIN^c_{SMAP}}$, $\mathrm{SM2RAIN^c_{SMOS}}$, $\mathrm{SM2RAIN^c_{ASCAT}}$ and $\mathrm{IMERG-ER^c}$ refer to the SM2RAIN products with climatological correction (using the calibration dataset)

.

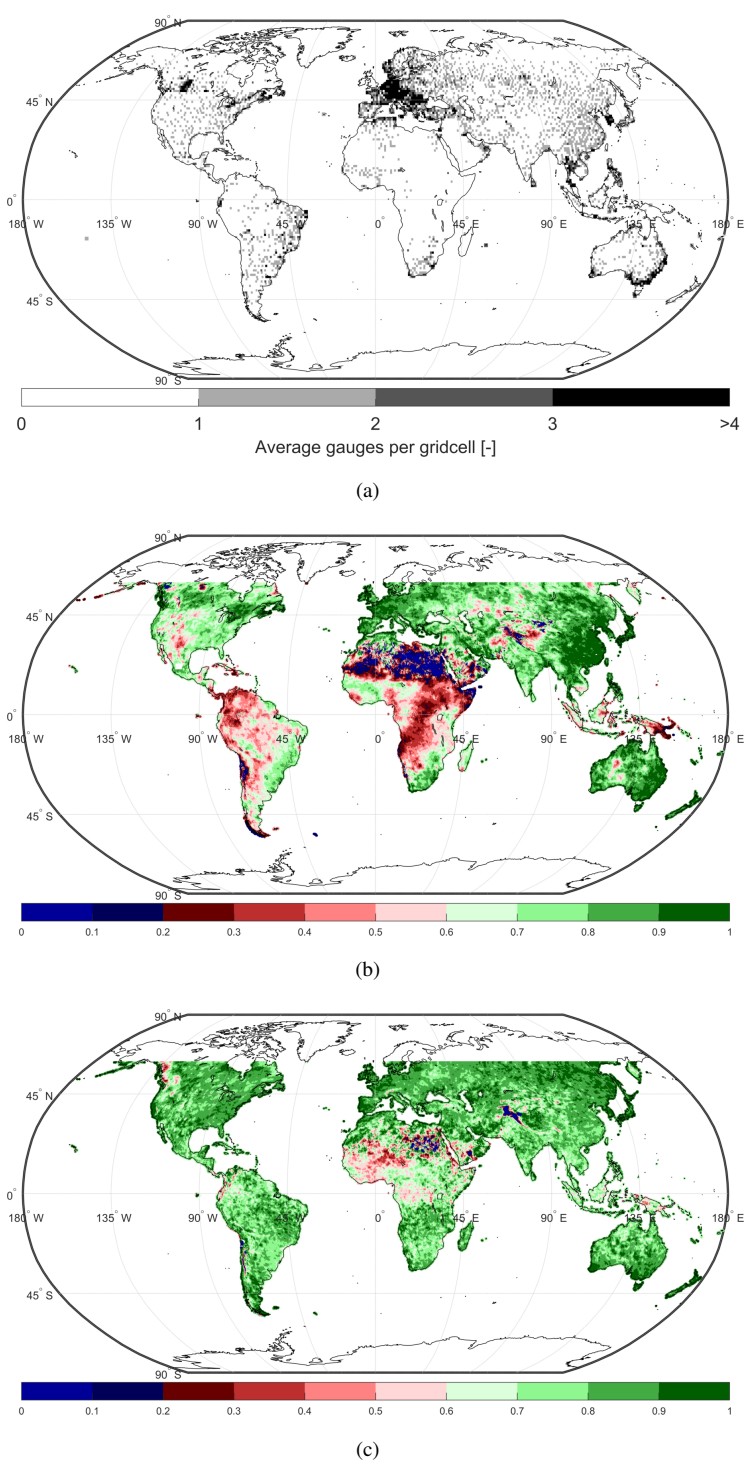

**Figure 2.** (a) Number of stations used for the GPCC First Guess 1.0 degree product for years 2015-2018. (b) TC correlation of the GPCC First Guess 1.0 in Africa during 2015-2018 using the triplet GPCC, ERA5, and SM2RAIN-ASCAT*. It can be seen that lower correlation areas closely match with low station density areas. (c) TC correlation of the ERA5 reanalysis during 2015-2018 GPCC, ERA5, and SM2RAIN-ASCAT*.



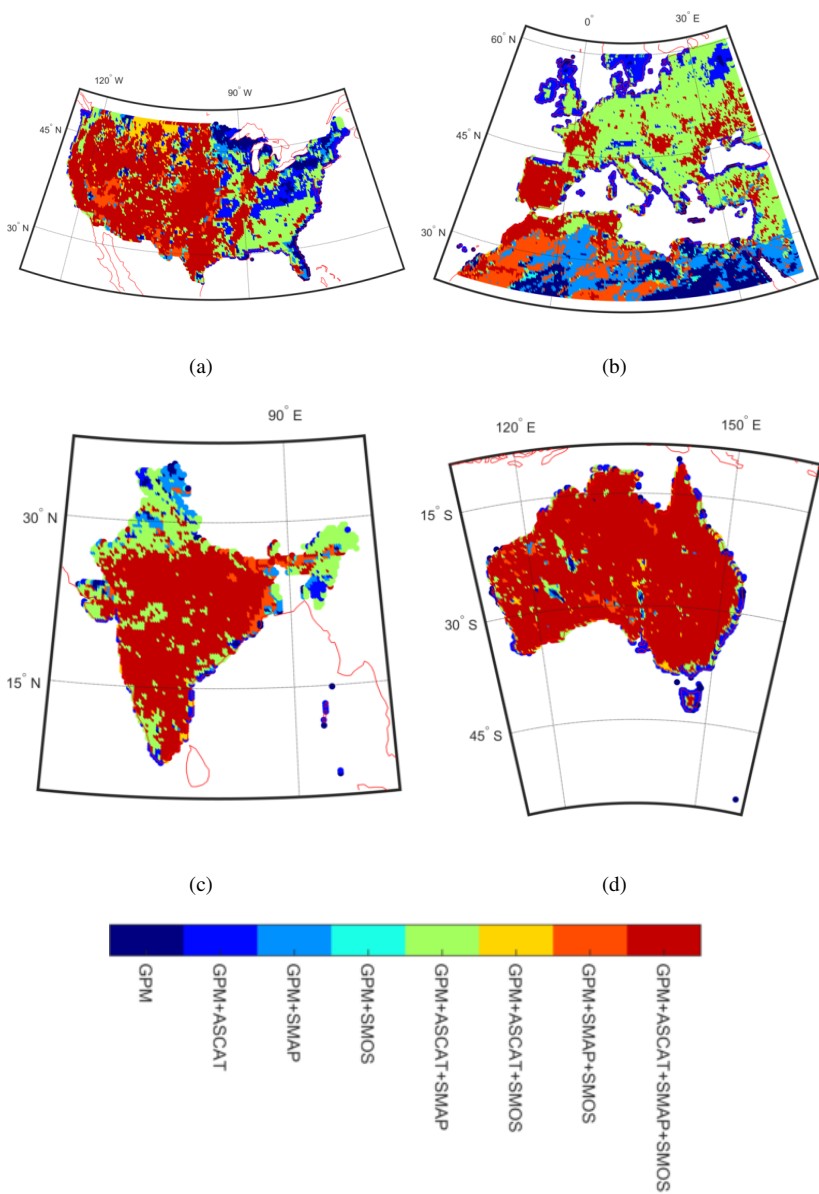

**Figure 3.** Products used in the integration over CONUS (a), Europe (b) and India (c) and Australia (d). The results refer to 2015-2017 period.



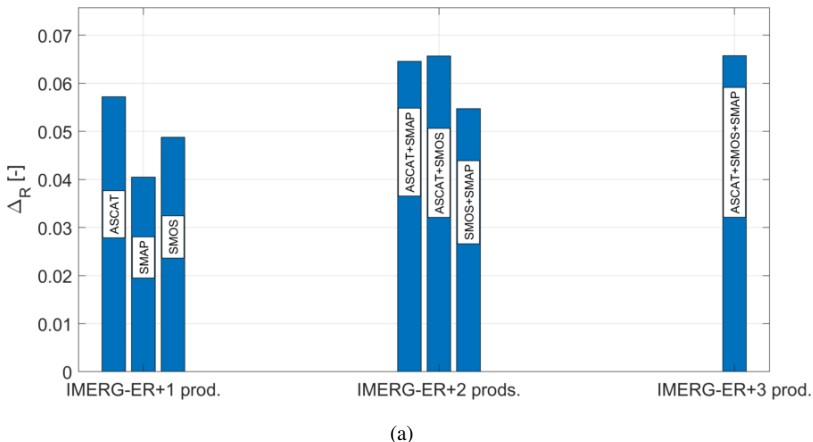

(a)

**Figure 4.** Increments obtained in terms of correlation against AWAP after the inclusion of more satellite soil moisture products via SM2RAIN in the integration scheme.

**Figure 5.** Percentage differences in correlation ($\Delta_R$ [%], panels a, b, c and d top) and in root mean square error ($\Delta_{RMSE}$ [%], panels f, g, h and i bottom) between the integrated product $P_{R+SM}$ and the IMERG Early Run product (IMERG-ER) over CONUS, Europe, India and Australia. Period 2015-2017. Grey areas represent the masked areas based on what described in Section 3.4.3.

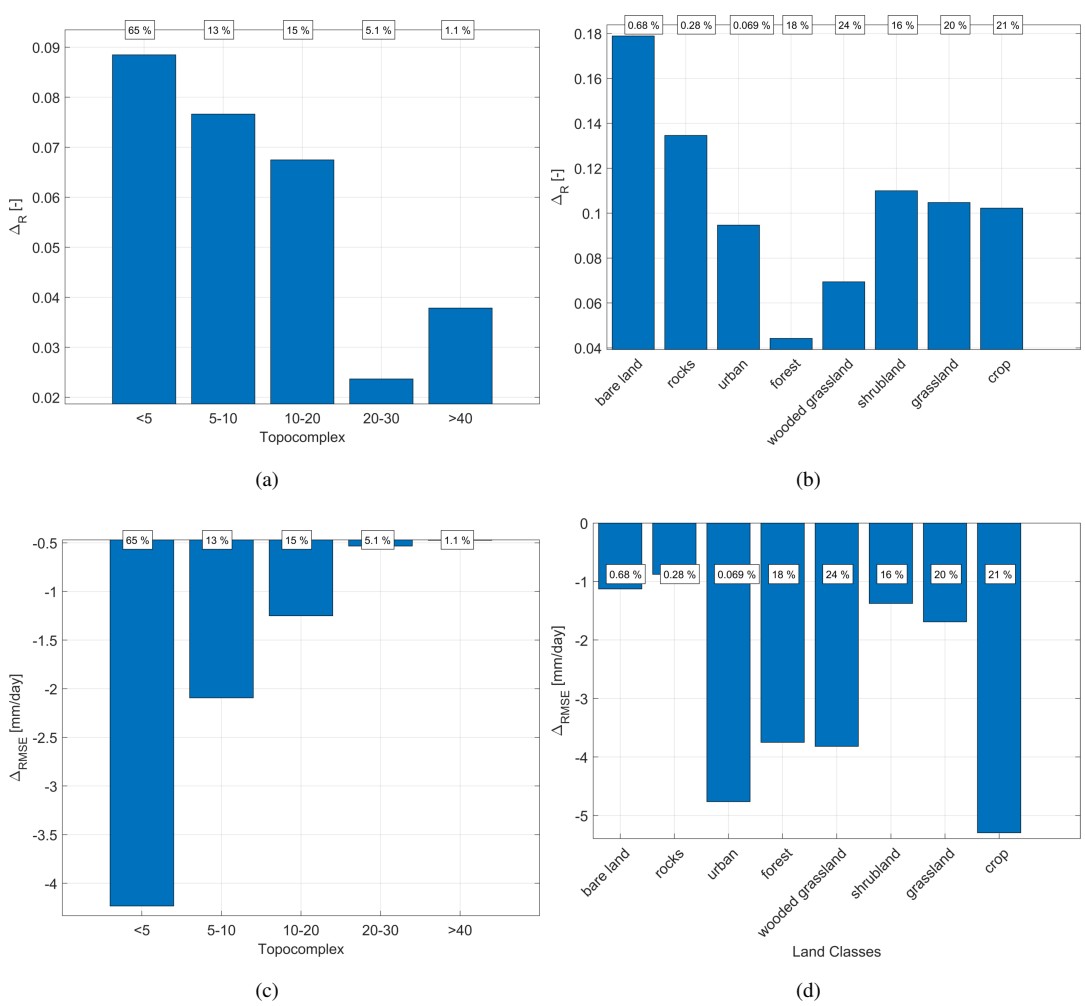

**Figure 6.** Difference in correlation ($\Delta_R$) and in root mean square error ($\Delta_{RMSE}$) between the integrated product $P_{R+SM}$ and IMERG-ER as a function of the topographic complexity (panels a and c) and as a function of the Land Cover type (panels b and d) over CONUS. The results refer to 2015-2017 period. The text boxes on the top show the percentage of the area occupied by the specific topographic complexity or land cover type.





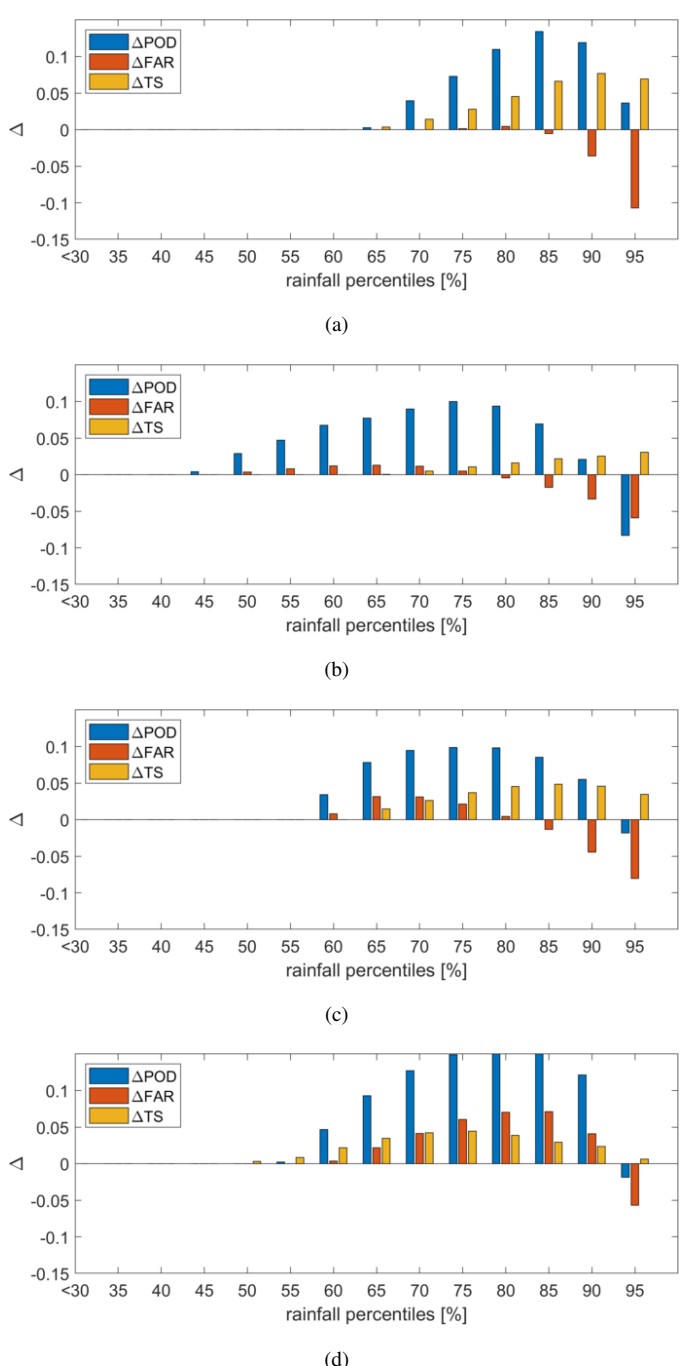

**Figure 7.** Difference in categorical indices (Probability of detection, POD, False Alarm Ratio, FAR and and Threat Score, TS) between the integrated product $P_{R+SM}$ and the IMERG-ER as a function of the rainfall classes for Australia (a), CONUS (b), Europe (c) and India (d). Period 2015-2017.





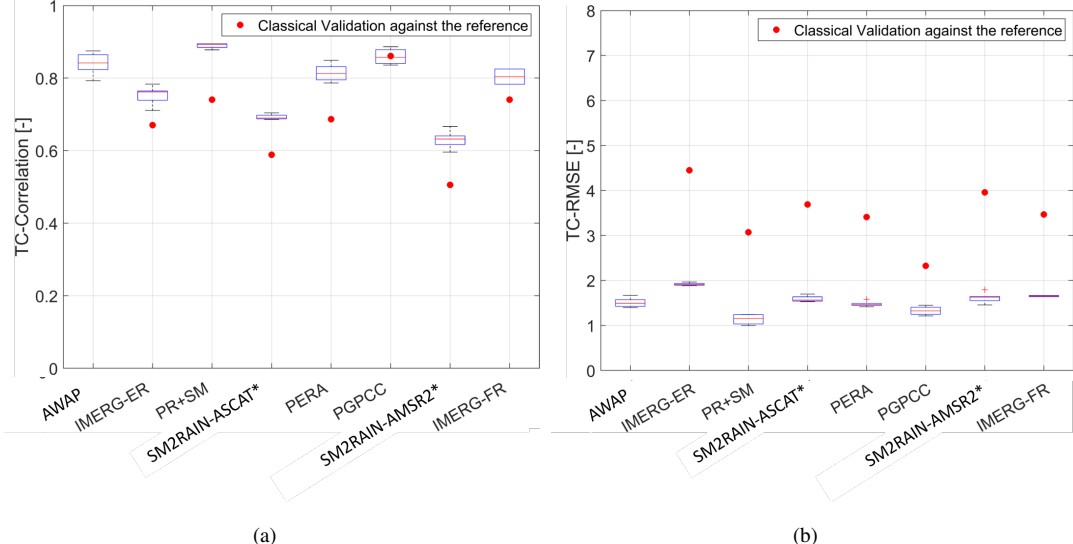

(a)                                                 (b)

**Figure 8.** Box plots of TC correlation $R_{TC}$ (a) and root mean squared error $RMSE_{TC}$ (b) obtained in Australia by considering the different outcomes of the 15 triplets compared with the classical correlation and RMSE scores (red dots) obtained by comparing the different rainfall products against AWAP.





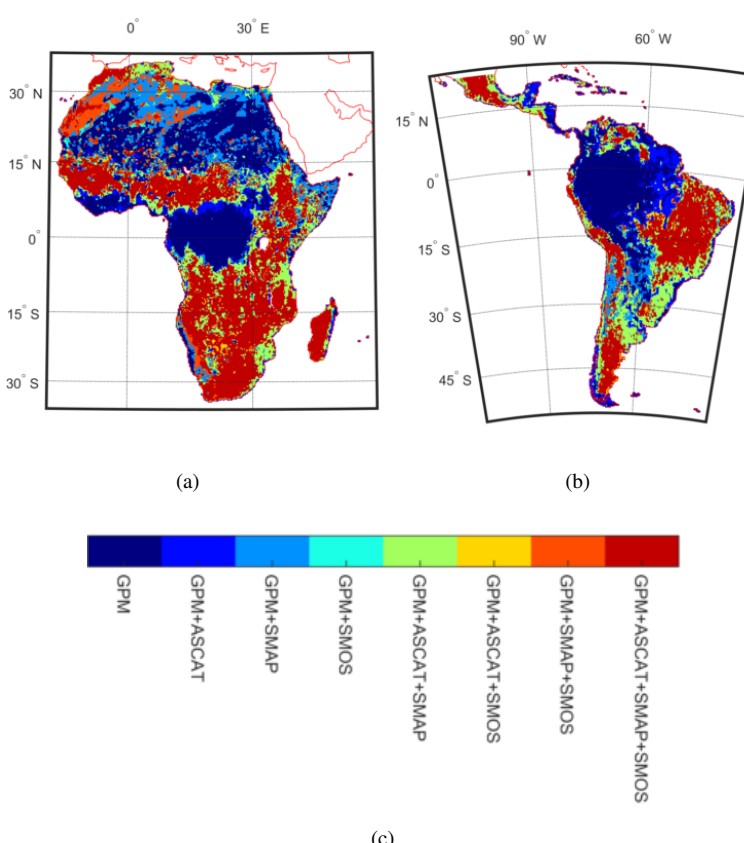

**Figure 9.** Products used to integrate IMERG-ER with SM2RAIN products derived from the setup during the calibration period in South America. When low correlation was found between the reference dataset (ie., GPCC) and the SM2RAIN product, the latter was excluded from the analysis and only IMERG-ER was retained.

**Figure 10.** Triple collocation squared correlation ($R^2_{TC}$, left) and Root Mean Square error ($RMSE_{TC}$, right) in mm/day for (top to down) IMERG Early run (IMERG-ER, panels a and f), the integrated product (IMERG early run + SM2RAIN applied to ASCAT, SMAP and SMOS, ($P_{R+SM}$, panels b and g), the Global Precipitation Climatology Center product (GPCC, panels c and h) and the reanalysis product ERA5 (ERA5, panels d and i). Gray areas represent the committed area of ASCAT which we excluded from the analysis. The results refer to the validation period (i.e., 2018). Grey areas represent the masked areas based on what described in Section 3.4.3.

**Figure 11.** As in Figure 10 but for South America.



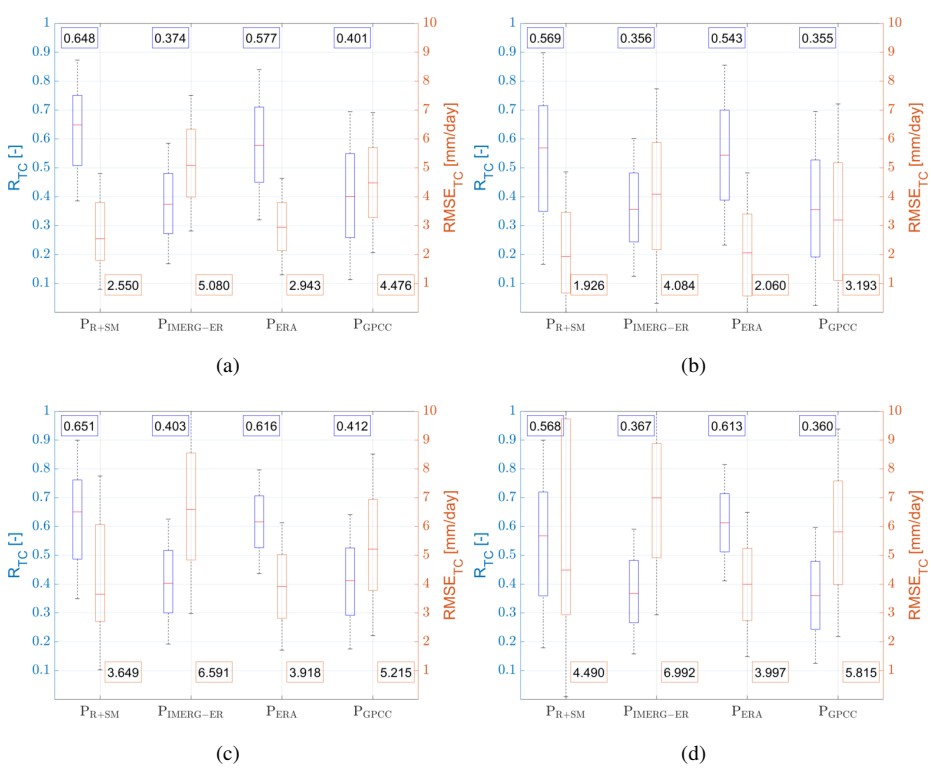

**Figure 12.** Box plots of Triple collocation squared correlation ($R_T^2$, left axis in blue) and Root Mean Square error ($RMSE_{TC}$, right axis in red) in mm/day obtained during validation period (i.e., 2018) in Africa over the committed area mask (a) and over all the study area (b). (c) and (d) refer to the same results but in South America. The box plot refer to the $25^{th}$ and $75^{th}$ percentiles while the whiskers refer the minimum and maximum values.





**Table 1.** Contingency table commonly used for characterizing detection errors of precipitation products.

|  | $R_{prod} \geq th$ | $R_{prod} < th$ |
|---|---|---|
| $R_{ref} \geq th$ | H | M |
| $R_{ref} < th$ | F | Z |





**Table 2.** Triple collocation correlation obtained by using triplet A) $P_{GPCC} - IMERG - ER - P_{SM2R-ASCAT}$, B) $P_{GPCC} - IMERG - ER - P_{ERA}$, C) $P_{GPCC} - P_{ERA} - P_{SM2R-ASCAT}$ for the period 2015-2018.

|  | A | B | C |
|---|---|---|---|
| GPCC | 0.7079 | 0.7144 | 0.6976 |
| ERA5 | - | 0.8136 | 0.8262 |
| IMERG-ER | 0.6671 | 0.6558 | |
| SM2RAIN-ASCAT | 0.7032 | - | 0.6791 |





**Table 3.** Correlation (R), root mean square error (RMSE) and daily bias (BIAS) obtained with the comparison of the different rainfall products against gauge-based AWAP (Australia), Stage IV (CONUS, gauge+radar), E-OBS (Europe) and IMD (India) during the period 2015-2017. $P_{R+SM}$ refers to the integrated product. Light blue highlighting refer to short-latency products.

| | | Australia | CONUS | Europe | India |
|---|---|---|---|---|---|
| **R [-]** | $P_{R+SM}$ | 0.770 | **0.705** | 0.679 | **0.740** |
| | IMERG-ER | 0.704 | 0.604 | 0.563 | 0.703 |
| | IMERG-FR | 0.767 | 0.665 | 0.630 | 0.737 |
| | GPCC | **0.878** | 0.696 | **0.898** | 0.595 |
| | ERA5 | 0.720 | 0.647 | 0.699 | 0.603 |
| **RMSE [mm/day]** | $P_{R+SM}$ | 3.043 | 3.562 | 3.016 | **5.074** |
| | IMERG-ER | 4.520 | 6.381 | 5.474 | 6.100 |
| | IMERG-FR | 3.509 | 4.689 | 4.542 | 6.142 |
| | GPCC | **2.306** | **3.446** | **1.751** | 6.669 |
| | ERA5 | 3.330 | 4.027 | 2.888 | 6.867 |
| **BIAS [mm/day]** | $P_{R+SM}$ | 0.090 | 0.135 | -0.035 | -0.238 |
| | IMERG-ER | -0.238 | -0.195 | -0.394 | -0.067 |
| | IMERG-FR | -0.076 | -0.008 | -0.457 | -0.129 |
| | GPCC | **0.002** | **0.072** | -0.118 | **-0.047** |
| | ERA5 | 0.087 | 0.129 | **-0.031** | -0.237 |