# Peer review of "A daily/25km short-latency rainfall product for data scarce regions based on the integration of the GPM IMERG Early Run with multiple satellite soil moisture products"

_Hydrology and Earth System Sciences, 2019_

## Referee Comment (RC1) · Anonymous Referee #1 · 25 Oct 2019

The paper introduces a potentially useful precipitation product and is overall quite well-written. However, some serious issues need to be resolved before it can be published.

"we minimised the daily root mean squared error (RMSE) between the SM2RAIN rainfall applied to the specific SM product and YREF during 2015-2017." This is problematic because of the noisy nature and highly skewed distribution of precipitation. ERA5 already underestimates precipitation peaks, and using this approach, the obtained SM2RAIN estimates will underestimate precipitation peaks even more. The Kling-Gupta Efficiency is probably a better choice as it accounts for the variability.

[Figure]

"The final product is then composed of multiple rainfall datasets weighed according to Eq. 6." An averaging scheme like this causes underestimation of peaks and introduces spurious drizzle. I realize that zero values of IMERG were kept, but this does not eliminate all spurious drizzle issue. It will, however, probably introduce a spurious discontinuity in the precipitation distribution...

"The continuous scores were the Pearson correlation coefficient (R), the Root Mean Squared Error (RMSE), and the additive bias (BIAS)." The RMSE statistic should not be used at the daily time scale because it yields "better" values for datasets which underestimate precipitation peaks (such as SM2RAIN and the dataset introduced here). The KGE (with its three independent components) is probably a better choice.

Overall, I think the authors should remove the RMSE from the evaluation and introduce metrics that evaluate the low and high tails of the precipitation distribution of the new product. Any issues revealed using these new metrics should be highlighted in the abstract.

"Note that, based on this choice, the integrated product is totally independent upon rain gauges" This not true as ERA5 assimilates precipitation gauge observations.

"ERA5, which provides full coverage and generally homogeneous performance all over the world." Not sure I agree with this as atmospheric models tend to perform markedly worse in convection-dominated regions.

Page 10 line 27: Add "out" after "carried".

Figure 12: Can you add short titles to each subplot?

---

## Referee Comment (RC2) · Anonymous Referee #2 · 12 Nov 2019

The manuscript describes a novel approach to integrate multiple precipitation estimates from satellite soil moisture measurements into an existing precipitation data set. The validity of the approach is confirmed over six regions using triple collocation validation and cross-validation with rain-gauge measurements where available.

The manuscript is overall well written and mostly well structured. Title and abstract fit the topic and content. Data generated within the study is publicly available, the URL in the manuscript should be updated in order to work properly (omit ".XThcfHvOOUk"). The supplementary materials contain plots and tables that are addressed in the text

and support the conclusions.

Using the SM2RAIN approach to improve existing precipitation products is a logical step in terms of deriving short-latency precipitation data and of interest for the community. The underlying concepts of the manuscript (OLI, SM2RAIN, TC) are well established, original contributions are summarized in a comprehensible way and referenced properly. More details on why the OLI method is preferred in the combination step over other approaches should be given. The chosen calibration data set $Y_{REF}$ within the combination step should show "homogeneous performance in space and time" globally, yet the chosen ERA5 reanalysis is likely not to provide this. ERA5 was found the best fitting of three candidates, yet potential issues of reanalysis products should be discussed and the (in)dependence of ERA5 from the used satellite data resp. raingauge data should be addressed thoroughly.

Some background for the chosen threshold of $R < 0.4$ between SM2RAIN and reference data to perform OLI should be given.

The "classical" validation part and the "assessment of TC validity" could be shorter or it should be explained, why extensive verification of the TC approach within this study was found to be necessary. As referenced in the manuscript, TC has been used to validate precipitation from satellite SM in a previous study (Massari et al., 2017).

Considering the short calibration/validation period, the potential impact of climate patterns or their absence (e.g. due to ENSO) on the calibration process should be discussed.

The increase in FAR in Fig. 7 needs more explanation.

The measures (median) that are shown in the (bar) plots should be defined in the plots or the caption (also for the tables). Areas that bar plots refer to are not always clear from the figures alone (Fig. 4), box plots resp. tables instead of bar plots would provide more information resp. improve readability/comparability. Information on what box

edges, whiskers represent in Fig.8 are especially necessary, to show that the impact of (single) outlier triples is not omitted in the plots.

P1, L5: "they are" instead of "they're" P4, L25: "Metop" instead of "METOP" P5, L12: A.M. - AM consistency P6, L27: "within" instead of "wihtin" P6, L29: "whereas suffer" missing words P8, L3: "weighting" instead of "weighing" P8, Eq. (5) : missing superscript "1" P11, L3: "satisfy" instead of "satisfies" P12, L23: duplicate section reference P20, L6: "targeted" instead of "target" and duplicate "(" in L5

---

## Author Comment (AC1) · 29 Nov 2019

The paper introduces a potentially useful precipitation product and is overall quite well written. However, some serious issues need to be resolved before it can be published. "we minimised the daily root mean squared error (RMSE) between the SM2RAIN rainfall applied to the specific SM product and YREF during 2015-2017." This is problematic because of the noisy nature and highly skewed distribution of precipitation. ERA5 already underestimates precipitation peaks, and using this approach, the obtained SM2RAIN estimates will underestimate precipitation peaks even more. The Kling-Gupta Efficiency is probably a better choice as it accounts for the variability.

**We thank the reviewer for their valuable suggestion. The reviewer is theoretically correct. KGE would theoretically provide a better dynamic range although our attempts to use it instead of RMSE did not always show these improvements. Moreover, by looking at Figure 7 (which presents the results obtained using RMSE) it can be seen a particular benefit to rainfall peaks with a reduction of FAR, an increase in POD and TS for percentiles larger than 90% which contrasts with what suggested by the reviewer.**
**In this study, we would rather ensure homogeneity among all the calibration steps, keeping in mind the results obtained here can be further improved by a better calibration using for instance KGE. We will clearly underline this issue in the revised version of the manuscript specifically in Section 3.3.**

"The final product is then composed of multiple rainfall datasets weighed according to Eq. 6." An averaging scheme like this causes underestimation of peaks and introduces spurious drizzle. I realize that zero values of IMERG were kept, but this does not eliminate all spurious drizzle issue. It will, however, probably introduce a spurious discontinuity in the precipitation distribution.

**We investigated this issue and found no detrimental effects of the current integration scheme on either low rainfall regimes (below the 50 percentile) or high peaks (rainfall percentiles larger than 90). By contrast we observed a slight increase of FAR in Australia and CONUS and little higher increments of this score for Europe and CONUS at medium/high rainfall regimes (60% up to 85%). Larger FARs were also accompanied by a significant increase in POD, which in turn determined the improvement in TS. As highlighted above, the change of the calibration score for SM2RAIN did not provide always a better behaviour and resulted in a POD decrease with overall smaller TS.**

"The continuous scores were the Pearson correlation coefficient (R), the Root Mean Squared Error (RMSE), and the additive bias (BIAS)." The RMSE statistic should not be used at the daily time scale because it yields "better" values for datasets which underestimate precipitation peaks (such as SM2RAIN and the dataset introduced here). The KGE (with its three independent components) is probably a better choice. Overall, I think the authors should remove the RMSE from the evaluation and introduce metrics that evaluate the low and high tails of the precipitation distribution of the new product. Any issues revealed using these new metrics should be highlighted in the abstract.

We thank the reviewer for this important comment. Any error-based metric like Means Squared Error (MSE) or KGE includes a correlation component, a variability component (often known as multiplicative bias or conditional bias), and an additive bias component. This can be demonstrated with some simple mathematical manipulations (see Murphy et al. 1988 and Gupta et al. 2009):

$$MSE = 2\sigma_s\sigma_o(1 - r) + (\sigma_s - \sigma_o)^2 + (\mu_s - \mu_o)^2$$

$$KGE = 1 - \sqrt{(r - 1)^2 + (\frac{\sigma_s}{\sigma_o} - 1)^2 + (\frac{\mu_s}{\mu_o} - 1)^2}$$

where $\sigma$ and $\mu$ refer to the standard deviation and the mean of the simulated "s" and the reference "o" time series whereas r is the correlation between them. So the three components are present also in the MSE or in its root version.
The notable difference of KGE with respect to MSE is the weight associated to the variability component which is larger for KGE with respect to MSE and the fact that it is a self consistent score as it varies from 0 to 1 (Gupta et al. 2009).

In the revised version of the manuscript we will provide the validation also in terms KGE score while the RMSE will be still maintained as:
1) many past studies are based on this metric and this facilitates the comparison of this work with them;
2) it is a physical error measure, therefore easier to relate to the actual physics of precipitation. (e.g. mm/day vs some fraction of KGE);
3) it can be compared against the results obtained via TC in a more meaningful way;

Gupta, H. V., Kling, H., Yilmaz, K. K., & Martinez, G. F. (2009). Decomposition of the mean squared error and NSE performance criteria: Implications for improving hydrological modelling. Journal of hydrology, 377(1-2), 80-91.

Murphy, A. H. (1988). Skill scores based on the mean square error and their relationships to the correlation coefficient. Monthly weather review, 116(12), 2417-2424.

"Note that, based on this choice, the integrated product is totally independent upon rain gauges" This not true as ERA5 assimilates precipitation gauge observations.

We agree with the reviewer ande have removed the sentence. However we want to highlight some points which we think would be interesting to discuss:

1) From the document "Operational global reanalysis: progress, future directions and synergies with NWP" which describes in details the development of ERA5 reanalysis dataset it is clear that after 2009 "rain rate" is ingested (Figure 17 panel k) in the reanalysis but it is not clear which rainfall information is ingested. To our knowledge only the NCEP Stage IV analysis which combines rain gauges and radars estimates are ingested into the analysis and ERA5 reanalysis but only in United States while no other gauge information is present in the two products outside this area (see also Lopez et al. 2011).

**Future developments will ingest radars information from OPERA ([https://www.ecmwf.int/en/elibrary/18765-operational-global-reanalysis-progress-future-directions-and-synergies-nwp](https://www.ecmwf.int/en/elibrary/18765-operational-global-reanalysis-progress-future-directions-and-synergies-nwp)) but currently this is not already done.**

2) **It is unlikely that rain gauges will be also ingested over data scarce regions as rain gauges are mostly absent over these regions (see Figure 2 in the paper).**
3) **Over CONUS we found relatively good correlations of ERA5 with Stage IV but also found it lower than the one obtained with our integrated product (see Table 3). This highlights that the integrated product is not really so dependant on the calibration dataset (i.e., ERA5) as highlighted also from the results in Figure 10 and 11.**

*Lopez, P. (2011). Direct 4D-Var assimilation of NCEP stage IV radar and gauge precipitation data at ECMWF. Monthly Weather Review, 139(7), 2098-2116.*

"ERA5, which provides full coverage and generally homogeneous performance all over the world." Not sure I agree with this as atmospheric models tend to perform markedly worse in convection-dominated regions.

**Our application of TC demonstrated that ERA5 is the best available calibration dataset among those selected, although it still suffers from uncertainty in convection dominated systems (i.e., Western Africa and Sahel see Figure 2). Despite this, the performance of the integrated product seems not to be too much impacted by its quality, as shown in Figure 10 (i.e.,compare results over Western Africa of ERA5 and the integrated product).**

Page 10 line 27: Add "out" after "carried".
**It will be corrected.**

Figure 12: Can you add short titles to each subplot?
**It will be done.**

---

## Author Comment (AC2) · 29 Nov 2019

The manuscript describes a novel approach to integrate multiple precipitation estimates from satellite soil moisture measurements into an existing precipitation data set. The validity of the approach is confirmed over six regions using triple collocation validation and cross-validation with rain-gauge measurements where available.

The manuscript is overall well written and mostly well structured. Title and abstract fit the topic and content. Data generated within the study is publicly available, the URL in the manuscript should be updated in order to work properly (omit ".XThcfHvOOUk"). The supplementary materials contain plots and tables that are addressed in the text and support the conclusions.

Using the SM2RAIN approach to improve existing precipitation products is a logical step in terms of deriving short-latency precipitation data and of interest for the community. The underlying concepts of the manuscript (OLI, SM2RAIN, TC) are well established, original contributions are summarized in a comprehensible way and referenced properly.

**We thank the reviewer for pointing this out and for the valuable suggestions. We will updated the link of the dataset.**

More details on why the OLI method is preferred in the combination step over other approaches should be given.

**Thanks for the comment. The ingestion of more SM2RAIN products implies that different satellite soil moisture products are used. Despite the potential benefits of one product with respect to another (i.e., active vs passive) the information we are ingesting is always soil moisture thus it is potentially redundant. The OLI method is particularly advantageous in this respect, as it accounts for both performance differences and error covariance between the rainfall products and is therefore insensitive to the addition of redundant information. We are aware that other more sophisticated methods exist (as for instance those based on data assimilation). However, it must be said that their better performance is not always guaranteed given the higher number of parameters upon which they rely. For instance, Brocca et al. (2017) found that simple integration methods performed equally well and in some cases better than data assimilation- based methods. We are also exploring new integration techniques (see Massari et al. 2019), but we think they are not mature enough for a continental scale application like this one. In the revised manuscript we will add more details in this respect.**

Brocca, Luca, et al. "Rainfall estimation by inverting SMOS soil moisture estimates: A comparison of different methods over Australia." Journal of Geophysical Research: Atmospheres 121.20 (2016): 12-062.

Massari, C., Maggioni, V., Barbetta, S., Brocca, L., Ciabatta, L., Camici, S., ... & Todini, E. (2019). Complementing near-real time satellite rainfall products with satellite soil moisture-derived rainfall through a Bayesian Inversion approach. Journal of Hydrology, 573, 341-351.

The chosen calibration data set YREF within the combination step should show "homogeneous performance in space and time" globally, yet the chosen ERA5 reanalysis is likely not to provide this. ERA5 was found the best fitting of three candidates, yet potential issues of reanalysis products should be discussed and the (in)dependence of ERA5 from the used satellite data resp. raingauge data should be addressed thoroughly.

**Thanks for this important comment. A perfect candidate is hard to find and even ERA5, which provided overall relatively good results, suffers from significant uncertainty in convective dominated systems (please refer to the reduced correlation in Western Africa and the Sahel in Figure 2). We considered three potential candidates and found that both GPCC and IMERG-FR strongly rely on rain gauge information, which for many regions may be scarce or absent altogether. This results in significant interpolation error of GPCC over Africa (Figure 2) and consequently potential detrimental effects of the gauge correction performed for IMERG-FR in the same regions. Thus, the problem of gauge absence seems much larger than the lower performance of the reanalysis dataset. Based on this consideration we selected ERA5 as calibration dataset.**
**We also tried to understand the impact of the quality of the calibration dataset on the integrated product. Based on the results reported in Table 3 and Figure 10, the integrated product seems slightly impacted by it (unless its performance is very bad) as:**
1) **in the regions where ERA5 performs relatively bad (Sahel and Western Africa) the integrated product still provides relatively good results ; and**
2) **the integrated product generally performs better than the calibration dataset in the cross-validation analysis indicating that its impact on the quality of the integrated product is small.**

**In the revised version of the manuscript we will discuss these aspects further trying to outline better the impact of Yref on the final results.**

Some background for the chosen threshold of R < 0:4 between SMt2RAIN and reference data to perform OLI should be given.
**This was simply a choice to exclude poor performance of SM2RAIN by accepting the risk of low correlations (<0.4) was due to the bad quality of ERA5. We performed some experiments over CONUS, Australia, Europe, and India and found that 0.4 is a reasonable threshold although its overall impact is very small and only limited to small regions (e.g., relatively high RFI regions for SMOS). We will add more details in the revised version of the manuscript trying to better explain the rationale behind the selection of this threshold.**

The "classical" validation part and the "assessment of TC validity" could be shorter or it should be explained, why extensive verification of the TC approach within this study was found to be necessary. As referenced in the manuscript, TC has been used to validate precipitation from satellite SM in a previous study (Massari et al., 2017).

**Massari et al. 2015 adopts a 1-deg resolution, so we thought it was important to validate the procedure at a finer scale (o.25deg). We will move the TC validation in the supplementary material.**

Considering the short calibration/validation period, the potential impact of climate patterns or their absence (e.g. due to ENSO) on the calibration process should be discussed.

**We are aware of potential problems related to this issue but at the time of writing IMERG (and SMAP) were only available since 2014 (2015). Now a reprocessed version of the IMERG products is available back to 2000. This is a potential opportunity to re-calibrate the integrated products with available soil moisture on specific years (i.e., ASCAT from 2007 onward, SMOS from 2010 onward, and SMAP from 2015 onward). We want to highlight however that this study shows already great potential in merging soil moisture-based rainfall observations with IMERG-ER and any limitation found at this stage (not directly related to the product development itself) represents additional room of improvement.**
**In the revised version of the manuscript specific sections/paragraphs will be devoted to the impact of the choice of a different calibration dataset on the final results.**

The increase in FAR in Fig. 7 needs more explanation.

**We observed an increasing trend in FAR from 60 to 80 percentiles and a drop above 85-90. This means that medium-high rainfall classes are increased more than necessary while high rainfall intensity are rightly reduced. While these increments in FAR are very small for Australia and CONUS, they are not negligible in India and Europe. Soil moisture-based rainfall has a tendency to generally provide higher FAR (deterioration) and higher POD (improvement) than IMERG-ER. When merged, SM2RAIN and IMERG-ER, result in a small increase of FAR and a significant increase of POD with an overall improvement of TS. Although the integration is overall optimal (TS increases), the POD increment -not surprisingly - causes an increase in FAR and viceversa . This is a classical multi-objective calibration exercise where, for the choice of the best configuration, one has to operate with a specific application in mind.**
**In addition, we found that the higher FAR for SM2RAIN datasets could be reduced by changing the calibration score to KGE, which is less affected by conditional biases , however, this also reduced the POD with an overall lower TS.**

**In the revised manuscript we will expand the discussion on these issues when discussing the figure and try to provide more insights and explanations on this issue.**

The measures (median) that are shown in the (bar) plots should be defined in the plots or the caption (also for the tables).

**It will be done.**

Areas that bar plots refer to are not always clear from the figures alone (Fig. 4), box plots resp. tables instead of bar plots would provide more information resp. improve readability/comparability. Information on what box edges, whiskers represent in Fig.8 are especially necessary, to show that the impact of (single) outlier triples is not omitted in the plots.

**We will try to use tables/box plots instead of bars in Figure 4. Also, all the missing information in the caption will be added.**

P1, L5: "they are" instead of "they're" P4, L25: "Metop" instead of "METOP" P5, L12: A.M. - AM consistency P6, L27: "within" instead of "wihtin" P6, L29: "whereas suffer" missing words P8, L3: "weighting" instead of "weighing" P8, Eq. (5) : missing superscript "1" P11, L3: "satisfy" instead of "satisfies" P12, L23: duplicate section reference P20, L6: "targeted" instead of "target" and duplicate "(" in L5

**Thanks a lot. We will correct all these typos.**

---

## Author Response (AR1)

**General comments**

This is the revised version of the manuscript "A daily/25km short-latency rainfall product for data scarce regions based on the integration of the GPM IMERG Early Run with multiple satellite soil moisture products" of Massari et al.

The manuscript has been extensively revised to address the reviewers' comments. We made our best effort to clarify all the issues raised and hope they are now satisfactorily addressed. In particular, we have included new scores to validate the products (i.e., KGE and variability ratio) as suggested by Reviewer 1 and better discussed many aspects related to the results and to the calibration dataset as suggested by Reviewer 2. The assessment of the validity of the TC analysis has been moved to the supplementary material as requested. We would like to thank the reviewers for their suggestions, which have greatly improved the quality of our manuscript.

The **answers** to the reviewers' comments are provided in **bold** while the ***manuscript text changes*** are reported in ***red*** with the indication of the lines where these changes are located.

**Answers to the comments of Reviewer 1**

1. *The paper introduces a potentially useful precipitation product and is overall quite well written. However, some serious issues need to be resolved before it can be published. "we minimised the daily root mean squared error (RMSE) between the SM2RAIN rainfall applied to the specific SM product and YREF during 2015-2017." This is problematic because of the noisy nature and highly skewed distribution of precipitation. ERA5 already underestimates precipitation peaks, and using this approach, the obtained SM2RAIN estimates will underestimate precipitation peaks even more. The Kling-Gupta Efficiency is probably a better choice as it accounts for the variability.*

**We thank the reviewer for their valuable suggestion. KGE would theoretically provide a better dynamic range although our attempts to use it instead of RMSE did not always show these improvements. Moreover, Figure 7 (which presents the results obtained using RMSE) shows some benefit to rainfall peaks with a reduction of FAR, an increase in POD and TS for percentiles larger than 90%, which is not in line with the reviewer's recommendation. In this study, we would rather ensure homogeneity among all the calibration steps, keeping in mind that results obtained here can be further improved by a better calibration using, for instance, KGE.**
**We have underlined this aspect in the revised manuscript (see lines 4-9 p. 10):**

[revised manuscript text omitted]

*5 ...”*

2. *"The final product is then composed of multiple rainfall datasets weighed according to Eq. 6." An averaging scheme like this causes underestimation of peaks and introduces spurious drizzle. I realize that zero values of IMERG were kept, but this does not eliminate all spurious drizzle issue. It will, however, probably introduce a spurious discontinuity in the precipitation distribution.*

We investigated this issue and found no significant detrimental effects of the current integration scheme on either low rainfall regimes (below the 50 percentile) or high peaks (rainfall percentiles larger than 90, see Figure 7 of the revised manuscript). By contrast, we observed a slight increase of FAR in Australia and CONUS and little higher increments of this score for Europe and CONUS at medium/high rainfall regimes (60% up to 85%). Larger FARs were also accompanied by a significant increase in POD, which in turn determined the improvement in TS.

In addition, to have an idea of how much the rainfall distribution of $P_{R+SM}$ , IMERG-ER, IMERG-FR, ERA5 and GPCC were similar to the benchmark, we compared the first four moments of their distributions (mean, variance, skewness and kurtosis) against those of the benchmark (pixel per pixel) and displayed them in four scatterplots each one representing a statistical moment. Optimal results for the scatterplot should lay on the 1:1 line.

We selected only Europe and CONUS to carry out this analysis to be sure to not have an impact of the interpolation used to align daily rainfall record of India and Australia to that of the 00:00 UTC. Indeed, while for Europe and CONUS daily rainfall data are provided at 00:00 UTC and recorded from 00:00 to 23:59, for AWAP and IMD (in Australia and India respectively) an interpolation was required to intercompare the products due to the different definition of the daily rainfall (from 9 AM) and the recording time. (local vs 00:00 UTC).

Figure R1 and Figure R2 show that every product is characterized by its own rainfall distribution with some products providing better behaviour in terms of mean and variance (i.e., GPCC) and others providing a very different one (IMERG-ER). In particular, $P_{R+SM}$ behaves relatively well in comparison with long latency products and does not seem to present serious problems in the rainfall distribution (at least is not much different from the others).

[Figure]

Figure R1. Scatterplot of the Mean (μ), variance (σ), skewness (w) and kurtosis (κ) associated to each pixel of PR+SM , IMERG-ER, IMERG-FR, ERA5 and GPCC and E-OBS dataset in Europe.

[Figure]

Figure R2. Scatterplot of the Mean (μ), variance (σ), skewness (w) and kurtosis (κ) associated to each pixel of PR+SM, IMERG-ER, IMERG-FR, ERA5 and GPCC and STAGE-IV dataset over CONUS.

3. *"The continuous scores were the Pearson correlation coefficient (R), the Root Mean Squared Error (RMSE), and the additive bias (BIAS)." The RMSE statistic should not be used at the daily time scale because it yields "better" values for datasets which underestimate precipitation peaks (such as SM2RAIN and the dataset introduced here). The KGE (with its three independent components) is probably a better choice. Overall, I think the authors should remove the RMSE from the evaluation and introduce metrics that evaluate the low and high tails of the precipitation distribution of the new product. Any issues revealed using these new metrics should be highlighted in the abstract.*

**We thank the reviewer for this comment. Any error-based metric like Means Squared Error (MSE) or KGE includes a correlation component, a variability component (often known as multiplicative bias or conditional bias), and an additive bias component. This can be demonstrated with some simple mathematical manipulations (see Murphy et al. 1988 and Gupta et al. 2009):**

$$MSE = 2\sigma_s\sigma_o(1 - r) + (\sigma_s - \sigma_o)^2 + (\mu_s - \mu_o)^2$$

$$KGE = 1 - \sqrt{(r - 1)^2 + (\frac{\sigma_s}{\sigma_o} - 1)^2 + (\frac{\mu_s}{\mu_o} - 1)^2}$$

**where σ and μ refer to the standard deviation and the mean of the simulated "s" and the reference "o" time series whereas r is the correlation between them. So the three components are present also in the MSE or in its root version.**
**The notable difference of KGE with respect to MSE is the weight associated to the variability component which is larger for KGE with respect to MSE and the fact that it is a self-consistent score as it varies from 0 to 1 (Gupta et al. 2009).**

**In the revised version of the manuscript we have included KGE and the variability ratio scores (see our replies to the points above and its inclusion in the abstract, methods, results and discussion). However, we have maintained the RMSE as:**

1) many past studies are based on this metric and this facilitates the comparison of this work with them;
2) it is a physical error measure, therefore easier to relate to the actual physics of precipitation (e.g., mm/day vs some fraction of KGE);
3) it can be compared against the results obtained via TC in a more meaningful way.

*Gupta, H. V., Kling, H., Yilmaz, K. K., & Martinez, G. F. (2009). Decomposition of the mean squared error and NSE performance criteria: Implications for improving hydrological modelling. Journal of hydrology, 377(1-2), 80-91.*
*Murphy, A. H. (1988). Skill scores based on the mean square error and their relationships to the correlation coefficient. Monthly weather review, 116(12), 2417-2424.*

4. *"Note that, based on this choice, the integrated product is totally independent upon rain gauges" This not true as ERA5 assimilates precipitation gauge observations.*

**The sentence has changed at (Line 23-24 pag. 15):**

**"Note that, except CONUS where rain gauge information is ingested into ERA5 (Lopez, 2011), the integrated product is totally independent upon rain gauge."**

**In addition, please, consider the following points:**

1) **From the document "Operational global reanalysis: progress, future directions and synergies with NWP" which describes in details the development of ERA5 reanalysis dataset it is clear that after 2009 "rain rate" is ingested (Figure 17 panel k) in the reanalysis but it is not clear which rainfall information is ingested. To our knowledge only the NCEP Stage IV analysis which combines rain gauges and radars estimates are ingested into the analysis and ERA5 reanalysis but only in United States while no other gauge information is present in the two products outside this area (see also Lopez et al. 2011). Future developments will ingest radar information from OPERA (https://www.ecmwf.int/en/elibrary/18765-operational-global-reanalysis-progress-future-directions-and-synergies-nwp) but currently this is not already done.**
2) **It is unlikely that rain gauges will be also ingested over data scarce regions as rain gauges are mostly absent over these regions (see Figure 2 in the paper).**
3) **Over CONUS we found relatively good correlations of ERA5 with Stage IV but also found it lower than the one obtained with our integrated product (see Table 3). This highlights that the integrated product is not really so dependant on the calibration dataset (i.e., ERA5) as highlighted also from the results in Figure 10 and 11 and in our previous studies (e.g., Brocca et al., 2019)**

*Lopez, P. (2011). Direct 4D-Var assimilation of NCEP stage IV radar and gauge precipitation data at ECMWF. Monthly Weather Review, 139(7), 2098-2116.*

5. *"ERA5, which provides full coverage and generally homogeneous performance all over the world." Not sure I agree with this as atmospheric models tend to perform markedly worse in convection-dominated regions.*

**Our application of TC demonstrated that ERA5 is the best available calibration dataset among those selected, although it still suffers from uncertainty in convection dominated systems (i.e., Western Africa and Sahel see Figure 2 of the revised manuscript). Despite this, the performance of the integrated product seems not to be too much impacted by its quality, as shown in Figure 10 (i.e.,compare results over Western Africa of ERA5 and the integrated product).**

**We have added some discussion on this issue at lines 17-22 pag. 15:**

*"This selection does not guarantee optimal solutions, but it is the best we can do with the available datasets considering that other potential candidates can be affected from other/similar issues, which could result in a very different global precipitation estimate (Herold et al., 2016). The solution to this problem is not straightforward, but a possible way forward would be the integration of GPCC and ERA5 or the use of available integrated products (Beck et al. 2017). The advantage of relying only on a single rainfall source (as to ensure homogeneity) however will be lost in that case."*

6.      Page 10 line 27: Add "out" after "carried".
**It has been corrected.**

7.      Figure 12: Can you add short titles to each subplot?
**It has been added.**

**Answers to the comments of Reviewer 2**

1.   The manuscript describes a novel approach to integrate multiple precipitation estimates from satellite soil moisture measurements into an existing precipitation data set. The validity of the approach is confirmed over six regions using triple collocation validation and cross-validation with rain-gauge measurements where available. The manuscript is overall well written and mostly well structured. Title and abstract fit the topic and content. Data generated within the study is publicly available, the URL in the manuscript should be updated in order to work properly (omit ".XThcfHvOOUk"). The supplementary materials contain plots and tables that are addressed in the text and support the conclusions. Using the SM2RAIN approach to improve existing precipitation products is a logical step in terms of deriving short-latency precipitation data and of interest for the community. The underlying concepts of the manuscript (OLI, SM2RAIN, TC) are well established, original contributions are summarized in a comprehensible way and referenced properly.

We thank the reviewer for pointing this out and for the valuable suggestions. We have updated the link.

2. More details on why the OLI method is preferred in the combination step over other approaches should be given.

Thanks for the comment. The ingestion of more SM2RAIN products implies that different satellite soil moisture products are used. Despite the potential benefits of one product with respect to another (i.e., active vs passive) the information we are ingesting is always soil moisture, thus it is potentially redundant. The OLI method is particularly advantageous in this respect, as it accounts for both performance differences and error covariance between the rainfall products and is therefore insensitive to the addition of redundant information. We are aware that other more sophisticated methods exist (as for instance those based on data assimilation). However, it must be said that their better performance is not always guaranteed given the higher number of parameters upon which they rely. For instance, Brocca et al. (2016) found that simple integration methods performed equally well and in some cases better than data assimilation-based methods. We are also exploring new integration techniques (see Massari et al. 2019), but we think they are not mature enough for a quasi-global scale application like this one.

In the revised manuscript we have added the following text in the OLI section at lines 26 pag. 8 to 4 pag. 9 to clarify this issue:

*"In addition, it is worth mentioning that the rainfall information brought from different SM2RAIN products to IMERG-ER is potentially redundant especially when the SM estimates from SMAP, ASCAT, and SMOS agree each other. The OLI method is particularly advantageous in this sense, as it accounts for both performance differences and error covariance between the rainfall products, and is therefore insensitive to the addition of redundant information. Other more sophisticated methods can be also applied, although there is no guarantee that such methods would lead to better results. For instance, Brocca et al. (2016) found that simple integration methods performed equally well and in some cases even better than more complex methods. Future developments will explore new and more complex integration techniques, as the one in Massari et al. (2019)."*

*Brocca, Luca, et al. "Rainfall estimation by inverting SMOS soil moisture estimates: A comparison of different methods over Australia." Journal of Geophysical Research: Atmospheres 121.20 (2016): 12-062.*

*Massari, C., Maggioni, V., Barbetta, S., Brocca, L., Ciabatta, L., Camici, S., ... & Todini, E. (2019). Complementing near-real time satellite rainfall products with satellite soil moisture-derived rainfall through a Bayesian Inversion approach. Journal of Hydrology, 573, 341-351.*

3. The chosen calibration data set YREF within the combination step should show "homogeneous performance in space and time" globally, yet the chosen ERA5

reanalysis is likely not to provide this. ERA5 was found the best fitting of three candidates, yet potential issues of reanalysis products should be discussed and the (in)dependence of ERA5 from the used satellite data resp. raingauge data should be addressed thoroughly.

**Thanks for this important comment. A perfect candidate is hard to find and even ERA5, which provided overall relatively good results, suffers from significant uncertainty in convective dominated systems (please refer to the reduced correlation in Western Africa and the Sahel in Figure 2). We considered three potential candidates and found that both GPCC and IMERG-FR strongly rely on rain gauge information, which for many regions may be scarce or absent altogether.**

**It has to note that the choice of ERA5 was not assumed a priori but was the logical result of the application of TC to global scale as discussed in Section 4.1. Results of TC indicate the superiority of ERA5 with respect to GPCC in many areas worldwide especially over Africa where the problem of rainfall interpolation error seems very significant (Figure 2). Nevertheless this step could be improved by integrating GPCC and ERA5 in a long-latency high accurate rainfall dataset to be used for calibrating the procedure proposed in this study, but this would require additional analyses which are out of the scope of this manuscript. Moreover, such an integrated product would likely provide less homogeneous rainfall estimates especially in terms of bias and thus a different rainfall distribution in space (see Figure R1 to the answer to reviewer 1).**

**We also tried to understand the impact of the quality of the calibration dataset on the integrated product. Based on the results reported in Table 3 and Figure 10, the integrated product seems to be slightly impacted by it (unless its performance is very bad) as:**

1) **in the regions where ERA5 performs relatively bad (Sahel and Western Africa), the integrated product still provides relatively good results ; and**
2) **the integrated product generally performs better than the calibration dataset in the cross-validation analysis indicating that its impact on the quality of the integrated product is not significant.**

**We have included an additional discussion to highlight these issues at lines 17-22 pag. 15:**

*"This selection does not guarantee optimal solutions, but it is the best we can do with the available datasets considering that other potential candidates can be affected from other/similar issues, which could result in a very different global precipitation estimate (Herold et al., 2016). The solution to this problem is not straightforward, but a possible way forward would be the integration of GPCC and ERA5 or the use of available integrated products (Beck et al. 2017). The advantage of relying only on a single rainfall source (as to ensure homogeneity) however will be lost in that case."*

4. Some background for the chosen threshold of R < 0.4 between SMt2RAIN and reference data to perform OLI should be given.

This was simply a choice to exclude poor performance of SM2RAIN by accepting the risk of low correlations (<0.4) was due to the bad quality of ERA5. We performed some experiments over CONUS, Australia, Europe, and India and found that 0.4 is a reasonable threshold although its overall impact is very small and only limited to small regions (e.g., relatively high RFI regions for SMOS).
We have added more explanations on this on the revised version of the manuscript at lines 8-14 pag. 11.

*"0.4 was set to exclude the poor performance of SM2RAIN products at such thresholds, which could potentially impact the overall quality of the integrated product. To select this value, we performed ad hoc experiments (not shown) over CONUS, Australia, Europe, and India and found 0.4 as a good compromise to exclude problematic areas like those impacted by high RFI in the SMOS SM product. However, its overall impact on the final results was found very small and only limited to some specific regions (e.g., high RFI, dense forests, and desert areas, which were already masked out by the validation mask)."*

5. The "classical" validation part and the "assessment of TC validity" could be shorter or it should be explained, why extensive verification of the TC approach within this study was found to be necessary. As referenced in the manuscript, TC has been used to validate precipitation from satellite SM in a previous study (Massari et al., 2017).

Massari et al. 2015 adopts a 1-deg resolution, so we thought it was important to validate the procedure at a finer scale (0.25-deg). We have now moved the Assessment of the TC validity in the supplementary information. This has been highlighted at lines 11-14 pag. 18:

*"Prior to the assessment of the rainfall products over Africa and South America with TC, we run TC analysis over AU, CONUS, EU and IN where R and RMSE scores obtained with the classical validation are available. Results are described in Appendix A of the Supplementary material and show that TC provides similar conclusions to a classical validation and can therefore be used as a robust validation tool over data scarce regions."*

6. Considering the short calibration/validation period, the potential impact of climate patterns or their absence (e.g. due to ENSO) on the calibration process should be discussed.

We are aware of potential problems related to this issue but at the time of writing IMERG (and SMAP) were only available since 2014 (2015). Now a reprocessed version of the IMERG products is available back to 2000. This is a potential opportunity to re-calibrate the integrated products with available soil moisture on specific years (i.e., ASCAT from 2007 onward, SMOS from 2010 onward, and SMAP from 2015 onward).

We want to highlight however that this study shows already great potential in merging soil moisture-based rainfall observations with IMERG-ER and any limitation found at this stage (not directly related to the product development itself) represents additional room of improvement.

In the revised version we have highlighted this issue in the discussion section. Now it reads (see lines 23-25 pag. 20):

*"The relatively short period of calibration has therefore potential impacts on the ability of the products to reproduce correct climate patterns. Thanks to the recent availability of the IMERG-ER from 2000 this aspect will be further investigated in the future versions of the product."*

7. The increase in FAR in Fig. 7 needs more explanation.

We observed an increasing trend in FAR from 60 to 80 percentiles and a drop above 85-90. This means that medium-high rainfall classes are increased more than necessary while high rainfall intensity are rightly reduced. While these increments in FAR are very small for Australia and CONUS, they are not negligible in India and Europe. Soil moisture-based rainfall has a tendency to generally provide higher FAR (deterioration) and higher POD (improvement) than IMERG-ER. When merged, SM2RAIN and IMERG-ER, result in a small increase of FAR and a significant increase of POD with an overall improvement of TS. Although the integration is overall optimal (TS increases), the POD increment - not surprisingly - causes an increase in FAR and viceversa. This is a classical multi-objective calibration exercise where, for the choice of the best configuration, one has to operate with a specific application in mind.
In addition, we found that the higher FAR for SM2RAIN datasets could be reduced by changing the calibration score to KGE, which is less affected by conditional biases, however, this also reduced the POD with an overall lower TS.

In the revised manuscript, we have expanded the discussion and modified the text (see lines 1-14 pag. 18)

*"After 50-60th percentiles, a significant increment of POD is evident for all the study regions, whereas the differences in FAR denote a deterioration from the 50th to 80th percentile across CONUS, EU, AU (very small), and in IN (much larger). The latter seems caused by more noisy satellite SM observations over India, which directly impact the quality of SM2RAIN estimates (causing higher FAR, see also Zhan et al., 2015 and Massari et al., 2019). This problem could be faced by de-noising satellite SM observations with methods similar to the ones proposed by Massari et al. (2017b) and Su et al. (2014, 2015), or by selecting a higher rainfall threshold below which only IMERG-ER is retained (i.e., larger than 1 mm selected above). The improvement in terms of FAR becomes significant for higher rainfall accumulations (i.e., 95th percentile). The overall improvement*

*is shown by the TS score, which is generally positive suggesting that the integrated product helps to improve IMERG-ER in terms of categorical scores especially for 70-90th percentiles."*

8. The measures (median) that are shown in the (bar) plots should be defined in the plots or the caption (also for the tables).

**It has been done.**

9. Areas that bar plots refer to are not always clear from the figures alone (Fig. 4), box plots resp. tables instead of bar plots would provide more information resp. improve readability/comparability. Information on what box edges, whiskers represent in Fig.8 are especially necessary, to show that the impact of (single) outlier triples is not omitted in the plots.

**We have modified Figure 4 which is now plotted as box plots (see below):**

[Figure]

Figure R3: Correlation increments obtained by ingesting ASCAT, SMOS and SMAP SM2RAIN-based rainfall estimates into IMERG-ER product. Values in bold inside the box plots refer to the median increments expressed in terms of percentage. The box plot refer to the 25th and 75th percentiles while the whiskers refer the minimum and maximum values. Outliers are not shown in the plot.

**Moreover, we have updated figure 8 caption which is now in the supplementary information (Figure A1 and A2).**

Minor points.

P1, L5: "they are" instead of "they're"
**It has been corrected.**

P4, L25: "Metop" instead of "METOP"
**It has been corrected.**

P5, L12:A.M. - AM consistency
**It has been corrected.**
P6, L27: "within" instead of "wihtin"
**It has been corrected.**

P6, L29: "whereas suffer" missing words
**It has been corrected.**

P8, L3: "weighting" instead of "weighing"
**It has been corrected.**

P8, Eq. (5) : missing superscript "1"
**It has been corrected.**

P11, L3: "satisfy" instead of "satisfies"
**It has been corrected**

P12, L23: duplicate section reference
**It has been corrected**

P20, L6: "targeted" instead of "target" and duplicate "(" in L5
**It has been corrected**

Riepilogo
1/28/2020 4:26:57 PM

I documenti presentano alcune differenze.

**Nuovo documento:**
SMOS_RAINFALL_HESSD_2revision
42 pagine (7.74 MB)
1/28/2020 4:26:45 PM
Utilizzato per la visualizzazione dei risultati.

**Documento precedente:**
SMOS_RAINFALL_HESSD_1revision
42 pagine (7.71 MB)
1/28/2020 4:26:45 PM

Inizio: la prima modifica è a pagina 1.

Nessuna pagina eliminata

**Come leggere questo rapporto**

L'evidenziazione indica una modifica.
Il testo barrato indica il contenuto eliminato.
▲ indica pagine modificate.
↔ indica pagine spostate.

[revised manuscript text omitted]